# NEURAL MECHANICS: SYMMETRY AND BROKEN CONSERVATION LAWS IN DEEP LEARNING DYNAMICS

**Daniel Kunin\*, Javier Sagastuy-Brena, Surya Ganguli, Daniel L.K. Yamins, Hidenori Tanaka\*†**

Stanford University
† Physics & Informatics Laboratories, NTT Research, Inc.

## ABSTRACT

Understanding the dynamics of neural network parameters during training is one of the key challenges in building a theoretical foundation for deep learning. A central obstacle is that the motion of a network in high-dimensional parameter space undergoes discrete finite steps along complex stochastic gradients derived from real-world datasets. We circumvent this obstacle through a unifying theoretical framework based on intrinsic symmetries embedded in a network's architecture that are present for *any* dataset. We show that any such symmetry imposes stringent geometric constraints on gradients and Hessians, leading to an associated conservation law in the continuous-time limit of stochastic gradient descent (SGD), akin to Noether's theorem in physics. We further show that finite learning rates used in practice can actually break these symmetry induced conservation laws. We apply tools from finite difference methods to derive *modified gradient flow*, a differential equation that better approximates the numerical trajectory taken by SGD at finite learning rates. We combine modified gradient flow with our framework of symmetries to derive exact integral expressions for the dynamics of certain parameter combinations. We empirically validate our analytic expressions for learning dynamics on VGG-16 trained on Tiny ImageNet. Overall, by exploiting symmetry, our work demonstrates that we can analytically describe the learning dynamics of various parameter combinations at finite learning rates and batch sizes for state of the art architectures trained on *any* dataset.

## 1 INTRODUCTION

Just like the fundamental laws of classical and quantum mechanics taught us how to control and optimize the physical world for engineering purposes, a better understanding of the laws governing neural network learning dynamics can have a profound impact on the optimization of artificial neural networks. This raises a foundational question: *what, if anything, can we quantitatively understand about the learning dynamics of large-scale, non-linear neural network models driven by real-world datasets and optimized via stochastic gradient descent with a finite batch size, learning rate, and with or without momentum?* In order to make headway on this extremely difficult question, existing works have made major simplifying assumptions on the network, such as restricting to identity activation functions Saxe et al. (2013), infinite width layers Jacot et al. (2018), or single hidden layers Saad & Solla (1995). Many of these works have also ignored the complexity introduced by stochasticity and discretization by only focusing on the learning dynamics under gradient flow. In the present work, we make the first step in an orthogonal direction. Rather than introducing unrealistic assumptions on the model or learning dynamics, we uncover restricted, but meaningful, combinations of parameters with simplified dynamics that can be solved exactly without introducing a major assumption (see Fig. 1). To find the parameter combinations, we use the lens of *symmetry* to show that if the training loss doesn't change under some transformation of the parameters, then the gradient and Hessian for those parameters have associated geometric constraints. We systematically apply this approach to modern neural networks to derive exact integral expressions and verify our predictions empirically on large scale models and datasets. We believe our work is the first step towards a foundational understanding

---

\* Equal contribution. Correspondence to kunin@stanford.edu & hidenori.tanaka@ntt-research.com

of neural network learning dynamics that is not based in simplifying assumptions, but rather the simplifying symmetries embedded in a network's architecture. Our main contributions are:

1. We leverage continuous differentiable symmetries in the loss to unify and generalize geometric constraints on neural network gradients and Hessians (section 3).
2. We prove that each of these differentiable symmetries has an associated conservation law under the learning dynamics of gradient flow (section 4).
3. We construct a more realistic continuous model for stochastic gradient descent by modeling weight decay, momentum, stochastic batches, and finite learning rates (section 5).
4. We show that under this more realistic model the conservation laws of gradient flow are broken, yielding simple ODEs governing the dynamics for the previously conserved parameter combinations (section 6).
5. We solve these ODEs to derive exact learning dynamics for the parameter combinations, which we validate empirically on VGG-16 trained on Tiny ImageNet with and without batch normalization (section 6).

## 2    RELATED WORK

The goal of this work is to construct a theoretical framework to better understand the learning dynamics of state-of-the-art neural networks trained on real-world datasets. Existing works have made progress towards this goal through major simplifying assumptions on the architecture and learning rule. Saxe et al. (2013; 2019) and Lampinen & Ganguli (2018) considered linear neural networks with specific orthogonal initializations, deriving exact solutions for the learning dynamics under gradient flow. The theoretical tractability of linear networks has further enabled analyses on the properties of loss landscapes Kawaguchi (2016), convergence Arora et al. (2018a); Du & Hu (2019), and implicit acceleration by overparameterization Arora et al. (2018b). Saad & Solla (1995) and Goldt et al. (2019) studied single hidden layer architectures with non-linearities in a student-teacher setup, deriving a set of complex ODEs describing the learning dynamics. Such shallow neural networks have also catalyzed recent major advances in understanding convergence

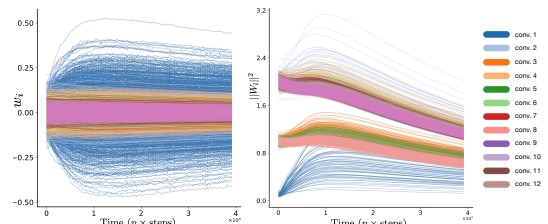

(a) Parameter Dynamics    (b) Neuron Dynamics

Figure 1: **Neuron level dynamics are simpler than parameter dynamics.** We plot the per-parameter dynamics (left) and per-channel squared Euclidean norm dynamics (right) for the convolutional layers of a VGG-16 model (with batch normalization) trained on Tiny ImageNet with SGD with learning rate $\eta = 0.1$, weight decay $\lambda = 10^{-4}$, and batch size $S = 256$. While the parameter dynamics are noisy and chaotic, the neuron dynamics are smooth and patterned.

properties of neural networks Du et al. (2018b); Mei et al. (2018). Jacot et al. (2018) considered infinitely wide neural networks with non-linearities, demonstrating that the network's prediction becomes linear in its parameters. This setting allows for an insightful mathematical formulation of the network's learning dynamics as a form of kernel regression where the kernel is defined by the initialization (though see also Fort et al. (2020)). Arora et al. (2019) extended these results to convolutional networks and Lee et al. (2019) demonstrated how this understanding also allows for predictions of parameter dynamics.

In the present work, we make the first step in an orthogonal direction. Instead of introducing unrealistic assumptions, we discover restricted combinations of parameters for which we can find exact solutions, as shown in Fig. 1. We make this fundamental contribution by constructing a framework harnessing the geometry of the loss shaped by symmetry and realistic continuous equations of learning.

**Geometry of the loss.** A wide range of literature has discussed constraints on gradients originating from specific architectural building blocks of networks. For the first part of our work, we simplify, unify, and generalize the literature through the lens of symmetry.

The earliest works understanding the importance of invariances in neural networks come from the loss landscape literature Baldi & Hornik (1989) and the characterization of critical points in the presence

of explicit regularization Kunin et al. (2019). More recent works have studied implicit regularization originating from linear Arora et al. (2018b) and homogeneous Du et al. (2018a) activations, finding that gradient geometry plays an important role in constraining the learning dynamics. A different line of research studying the generalization capacity of networks has noticed similar gradient structures Liang et al. (2019). Beyond theoretical studies, geometric properties of the gradient and Hessian have been applied to optimize Neyshabur et al. (2015), interpret Bach et al. (2015), and prune Tanaka et al. (2020) neural networks.

Gradient properties introduced by batch Ioffe & Szegedy (2015), weight Salimans & Kingma (2016) and layer Ba et al. (2016) normalization have been intensely studied. Van Laarhoven (2017); Zhang et al. (2018) showed that normalization layers are scale invariant, but have an implicit role in controlling the effective learning rate. Cho & Lee (2017); Hoffer et al. (2018); Chiley et al. (2019); Li & Arora (2019); Wan et al. (2020) have leveraged the scale invariance of batch normalization to understand geometric properties of the learning dynamics. Most recently, Li et al. (2020) studied the role of gradient noise in reconciling the empirical dynamics of batch normalization with the theoretical predictions given by continuous models of gradient descent.

**Equations of learning.** To make experimentally testable predictions on learning dynamics, we introduce a continuous model for stochastic gradient descent (SGD). There exists a large body of works studying this subject using stochastic differential equations (SDEs) in the continuous-time limit Mandt et al. (2015; 2017); Li et al. (2017); Smith & Le (2017); Chaudhari & Soatto (2018); Jastrzębski et al. (2017); Zhu et al. (2018); An et al. (2018). Each of these works involves making specific assumptions on the loss or the noise in order to derive stationary distributions. More careful treatment of stochasticity led to fluctuation dissipation relationships at steady state without such assumptions Yaida (2018). In our analysis, we apply a more recent approach Li et al. (2017); Barrett & Dherin (2020), inspired by finite difference methods, that augments SDE model with higher-order terms to account for the effect of a finite step size and curvature in the learning trajectory.

## 3 SYMMETRIES IN THE LOSS SHAPE GRADIENT AND HESSIAN GEOMETRIES

While we initialize neural networks randomly, their gradients and Hessians at all points in training, no matter the loss or dataset, obey certain geometric constraints. Some of these constraints have been noticed previously as a form of implicit regularization, while others have been leveraged algorithmically in applications from network pruning to interpretability. Remarkably, all these geometric constraints can be understood as consequences of numerous differentiable symmetries in the loss introduced by neural network architectures. A set of parameters observes a differentiable symmetry in the loss if the loss doesn't change under a certain differentiable transformation of these parameters. This invariance introduces associated geometric constraints on the gradient and Hessian.

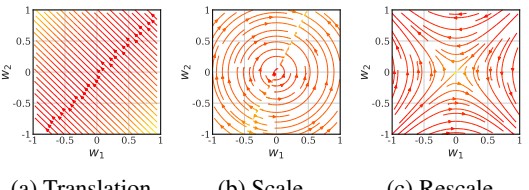

(a) Translation     (b) Scale     (c) Rescale

Figure 2: **Visualizing symmetry.** We visualize the vector fields associated with simple network components that have translation, scale, and rescale symmetry. In (a) we consider the vector field associated with a neuron $\sigma\left([w_1 \quad w_2]^\mathsf{T} x\right)$ where $\sigma$ is the softmax function. In (b) we consider the vector field associated with a neuron $\mathrm{BN}\left([w_1 \quad w_2][x_1 \quad x_2]^\mathsf{T}\right)$ where BN is the batch normalization function. In (c) we consider the vector field associated with a linear path $w_2 w_1 x$.

Consider a function $f(\theta)$ where $\theta \in \mathbb{R}^m$. This function possesses a differentiable symmetry if it is invariant under the differentiable action $\psi$ of a group $G$ on the parameter vector $\theta$, i.e., if $\theta \mapsto \psi(\theta, \alpha)$ where $\alpha \in G$, then $F(\theta, \alpha) = f(\psi(\theta, \alpha)) = f(\theta)$ for all $(\theta, \alpha)$. The existence of a symmetry enforces a geometric structure on the gradient, $\nabla F$. Evaluating the gradient at the identity element $I$ of $G$ (so that for all $\theta$, $\psi(\theta, I) = \theta$) yields the result,

$$\langle \nabla f, \partial_\alpha \psi|_{\alpha=I} \rangle = 0, \tag{1}$$

where $\langle\ ,\ \rangle$ denotes the inner product. This equality implies that the gradient $\nabla f$ is perpendicular to the vector field $\partial_\alpha \psi|_{\alpha=I}$ that generates the symmetry, for all $\theta$. The symmetry also enforces a geometric structure on the Hessian, $\mathbf{H}F$. Evaluating the Hessian at the identity element yields the

result,

$$\mathbf{H}f[\partial_\theta \psi|_{\alpha=I}]\partial_\alpha \psi|_{\alpha=I} + [\partial_\theta \partial_\alpha \psi|_{\alpha=I}]\nabla f = 0, \tag{2}$$

which constrains the Hessian $\mathbf{H}f$. See appendix A for the derivation of these properties and other geometric consequences of symmetry.

We will now consider the specific setting of a neural network parameterized by $\theta \in \mathbb{R}^m$, the training loss $\mathcal{L}(\theta)$, and three families of symmetries (translation, scale, and rescale) that commonly appear in modern network architectures.

**Translation symmetry.** Translation symmetry is defined by the group $\mathbb{R}$ and action $\theta \mapsto \theta + \alpha \mathbb{1}_\mathcal{A}$ where $\mathbb{1}_\mathcal{A}$ is the indicator vector for some subset $\mathcal{A}$ of the parameters $\{\theta_1, \ldots, \theta_m\}$. The loss function $\mathcal{L}$ thus possesses translation symmetry if $\mathcal{L}(\theta) = \mathcal{L}(\theta + \alpha \mathbb{1}_\mathcal{A})$ for all $\alpha \in \mathbb{R}$. Such a symmetry in turn implies the loss gradient $\partial_\theta \mathcal{L} = g$ is orthogonal to the indicator vector $\partial_\alpha \psi|_{\alpha=I} = \mathbb{1}_\mathcal{A}$,

$$\langle g, \mathbb{1}_\mathcal{A} \rangle = 0, \tag{3}$$

and that the Hessian matrix $H = \partial_\theta^2 \mathcal{L}$ has the indicator vector in its kernel,

$$H\mathbb{1}_\mathcal{A} = 0. \tag{4}$$

*Softmax function.* Any network using the softmax function gives rise to translation symmetry for the parameters immediately preceding the function. Let $z = Wx + b$ be the input to the softmax function such that $\sigma(z)_i = \frac{e^{z_i}}{\sum_j e^{z_j}}$. Notice that shifting any column of the weight matrix $w_i$ or the bias vector $b$ by a real constant has no effect on the output of the softmax as the shift factors from both the numerator and denominator canceling its effect. Thus, the loss function is invariant w.r.t. this translation, yielding the gradient constraints $\langle \frac{\partial \mathcal{L}}{\partial w_i}, \mathbb{1} \rangle = \langle \frac{\partial \mathcal{L}}{\partial b}, \mathbb{1} \rangle = 0$, visualized in Fig. 2 for the toy model where $w_i \in \mathbb{R}^2$.

**Scale symmetry.** Scale symmetry is defined by the group $\mathrm{GL}_1^+(\mathbb{R})$ and action $\theta \mapsto \alpha_\mathcal{A} \odot \theta$ where $\alpha_\mathcal{A} = \alpha \mathbb{1}_\mathcal{A} + \mathbb{1}_{\mathcal{A}^c}$. The loss function possesses scale symmetry if $\mathcal{L}(\theta) = \mathcal{L}(\alpha_\mathcal{A} \odot \theta)$ for all $\alpha \in \mathrm{GL}_1^+(\mathbb{R})$. This symmetry immediately implies the loss gradient is everywhere perpendicular to the parameter vector itself $\partial_\alpha \psi|_{\alpha=I} = \theta \odot \mathbb{1}_\mathcal{A} = \theta_\mathcal{A}$,

$$\langle g, \theta_\mathcal{A} \rangle = 0, \tag{5}$$

and relates to the Hessian matrix, where $[\partial_\alpha \partial_\theta \psi|_{\alpha=I}]\partial_\theta \mathcal{L} = \mathrm{diag}(\mathbb{1}_\mathcal{A})g = g_\mathcal{A}$, as

$$H\theta_\mathcal{A} + g_\mathcal{A} = 0. \tag{6}$$

*Batch normalization.* Batch normalization leads to scale invariance during training. Let $z = w^\mathsf{T} x + b$ be the input to a neuron with batch normalization such that $\mathrm{BN}(z) = \frac{z - \mathrm{E}[z]}{\sqrt{\mathrm{Var}(z)}}$ where $\mathrm{E}[z]$ is the sample mean and $\mathrm{Var}(z)$ is the sample variance given a batch of data. Notice that scaling $w$ and $b$ by a non-zero real constant has no effect on the output of the batch normalization as it factors from $z$, $\mathrm{E}[z]$, and $\mathrm{Var}(z)$ canceling its effect. Thus, these parameters observe scale symmetry in the loss and their gradients satisfy $\langle \frac{\partial \mathcal{L}}{\partial w}, w \rangle + \langle \frac{\partial \mathcal{L}}{\partial b}, b \rangle = 0$, as has been previously noted by Ioffe & Szegedy (2015); Van Laarhoven (2017), and visualized in Fig. 2 for the toy model where $w, b \in \mathbb{R}$.

**Rescale symmetry.** Rescale symmetry is defined by the group $\mathrm{GL}_1^+(\mathbb{R})$ and action $\theta \mapsto \alpha_{\mathcal{A}_1} \odot \alpha_{\mathcal{A}_2}^{-1} \odot \theta$ where $\mathcal{A}_1$ and $\mathcal{A}_2$ are two disjoint sets of parameters. The loss function possesses rescale symmetry if $\mathcal{L}(\theta) = \mathcal{L}(\alpha_{\mathcal{A}_1} \odot \alpha_{\mathcal{A}_2}^{-1} \odot \theta)$ for all $\alpha \in \mathrm{GL}_1^+(\mathbb{R})$. This symmetry immediately implies the loss gradient is everywhere perpendicular to the sign inverted parameter vector $\partial_\alpha \psi|_{\alpha=I} = \theta_{\mathcal{A}_1} - \theta_{\mathcal{A}_2} = \theta \odot (\mathbb{1}_{\mathcal{A}_1} - \mathbb{1}_{\mathcal{A}_2})$,

$$\langle g, \theta_{\mathcal{A}_1} - \theta_{\mathcal{A}_2} \rangle = 0 \tag{7}$$

and relates to the Hessian matrix, where $[\partial_\alpha \partial_\theta \psi|_{\alpha=I}]\partial_\theta \mathcal{L} = \mathrm{diag}(\mathbb{1}_{\mathcal{A}_1} - \mathbb{1}_{\mathcal{A}_2})g = g_{\mathcal{A}_1} - g_{\mathcal{A}_2}$, as

$$H(\theta_{\mathcal{A}_1} - \theta_{\mathcal{A}_2}) + g_{\mathcal{A}_1} - g_{\mathcal{A}_2} = 0. \tag{8}$$

*Homogeneous activation.* For networks with continuous, homogeneous activation functions $\phi(z) = \phi'(z)z$ (e.g. ReLU, Leaky ReLU, linear), this symmetry emerges at every hidden neuron by considering all incoming and outgoing parameters to the neuron. For example, consider a hidden neuron with ReLU activation $\phi(z) = \max\{0, z\}$, such that $w_2 \phi(w_1^\mathsf{T} x + b)$ is the computational path through this neuron. Scaling $w_1$ and $b$ by a real constant and $w_2$ by its inverse has

no effect on the computational path as the constants can be passed through the ReLU activation canceling their effects. Thus, these parameters observe rescale symmetry and their gradients satisfy $\left\langle \frac{\partial \mathcal{L}}{\partial w_1}, w_1 \right\rangle + \left\langle \frac{\partial \mathcal{L}}{\partial b}, b \right\rangle - \left\langle \frac{\partial \mathcal{L}}{\partial w_2}, w_2 \right\rangle = 0$, as has been previously noted by Du et al. (2018a); Liang et al. (2019); Tanaka et al. (2020), and visualized in Fig. 2 for the toy model where $w_1, w_2 \in \mathbb{R}$ and $b = 0$.

# 4 SYMMETRY LEADS TO CONSERVATION LAWS UNDER GRADIENT FLOW

We now explore how geometric constraints on gradients and Hessians, arising as a consequence of symmetry, impact the learning dynamics given by stochastic gradient descent (SGD). We will consider a model parameterized by $\theta$, a training dataset $\{x_1, \ldots, x_N\}$ of size $N$, and a training loss $\mathcal{L}(\theta) = \frac{1}{N} \sum_{i=1}^{N} \ell(\theta, x_i)$ with corresponding gradient $g(\theta) = \frac{\partial \mathcal{L}}{\partial \theta}$.

The gradient descent update with learning rate $\eta$ is $\theta^{(n+1)} = \theta^{(n)} - \eta g(\theta^{(n)})$, which is a forward Euler discretization with step size $\eta$ of the ordinary differential equation (ODE)

$$\frac{d\theta}{dt} = -g(\theta). \tag{9}$$

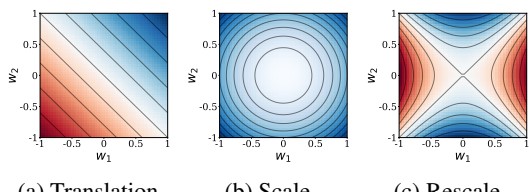

| (a) Translation | (b) Scale | (c) Rescale |

Figure 3: **Visualizing conservation.** Associated with each symmetry is a conserved quantity constraining the gradient flow dynamics to a surface. For translation symmetry (a) the flow is constrained to a hyperplane where the intercept is conserved. For scale symmetry (b) the flow is constrained to a sphere where the radius is conserved. For rescale symmetry (c) the flow is constrained to a hyperbola where the axes are conserved. The color represents the value of the conserved quantity, where blue is positive and red is negative, and the black lines are level sets.

In the limit as $\eta \to 0$, gradient descent exactly matches the dynamics of this ODE, which is commonly referred to as gradient flow Kushner & Yin (2003). Equipped with a continuous model for the learning dynamics, we now ask how do the dynamics interact with the geometric properties introduced by symmetries?

**Symmetry leads to conservation.** Strikingly similar to Noether's theorem, which describes a fundamental relationship between symmetry and conservation for physical systems governed by Lagrangian dynamics, here we show that symmetries of a network architecture have corresponding conserved quantities through training under gradient flow.

**Theorem 1.** *Symmetry and conservation laws in neural networks. Every differentiable symmetry $\psi(\alpha, \theta)$ of the loss that satisfies $\langle \theta, [\partial_\alpha \partial_\theta \psi|_{\alpha=I}] g(\theta) \rangle = 0$ has the corresponding conservation law,*

$$\frac{d}{dt} \langle \theta, \partial_\alpha \psi|_{\alpha=I} \rangle = 0, \tag{10}$$

*through learning under gradient flow.*

To prove Theorem 1, consider projecting the gradient flow learning dynamics in equation (9) onto the generator vector field $\partial_\alpha \psi|_{\alpha=I}$ associated with a symmetry. As shown in section 3, the gradient of the loss $g(\theta)$ is always perpendicular to the vector field $\partial_\alpha \psi|_{\alpha=I}$. Thus, the projection yields a differential equation, which can be simplified to equation (10). See appendix B for a complete derivation.

Application of this general theorem to the translation, scale, and rescale symmetries, identified in section 3, yields the following conservation law of learning,

$$\textbf{Translation:} \quad \langle \theta_{\mathcal{A}}(t), \mathbb{1} \rangle = \langle \theta_{\mathcal{A}}(0), \mathbb{1} \rangle \tag{11}$$

$$\textbf{Scale:} \quad |\theta_{\mathcal{A}}(t)|^2 = |\theta_{\mathcal{A}}(0)|^2 \tag{12}$$

$$\textbf{Rescale:} \quad |\theta_{\mathcal{A}_1}(t)|^2 - |\theta_{\mathcal{A}_2}(t)|^2 = |\theta_{\mathcal{A}_1}(0)|^2 - |\theta_{\mathcal{A}_2}(0)|^2 \tag{13}$$

Each of these equations define a conserved constant of learning through training. For parameters with translation symmetry, their sum ($\langle \theta_{\mathcal{A}}(t), \mathbb{1} \rangle$) is conserved, effectively constraining their dynamics to a hyperplane. For parameters with scale symmetry, their euclidean norm ($|\theta_{\mathcal{A}}(t)|^2$) is conserved,

effectively constraining their dynamics to a sphere Ioffe & Szegedy (2015); Van Laarhoven (2017). For parameters with rescale symmetry, their difference in squared euclidean norm ($|\theta_{\mathcal{A}_1}(t)|^2 - |\theta_{\mathcal{A}_2}(t)|^2$) is conserved, effectively constraining their dynamics to a hyperbola Du et al. (2018a). In Fig. 3 we visualize the level sets of these conserved quantities for the toy models discussed in Fig. 2.

## 5 A REALISTIC CONTINUOUS MODEL FOR STOCHASTIC GRADIENT DESCENT

In section 4 we combined the geometric constraints introduced by symmetries with gradient flow to derive conservation laws for simple combinations of parameters during training. However, empirically we know these laws are broken, as demonstrated in Fig. 1. What causes this discrepancy? Gradient flow is too simple of a continuous model for realistic SGD training. It fails to incorporate the effect of weight decay introduced by explicit regularization, momentum introduced by commonly used hyperparameters, stochasticity introduced by random batches, and discretization introduced by a finite learning rate. Here, we construct a more realistic continuous model for stochastic gradient descent.

**Modeling weight decay.** Explicit regularization through the addition of an $L_2$ penalty on the parameters, with regularization constant $\lambda$, is very common practice when training modern deep learning models. This is generally implemented not by modifying the training loss, but rather directly modifying the optimizer's update equation. For stochastic gradient descent, the result leads to the updated continuous model

$$\frac{d\theta}{dt} = -g(\theta) - \lambda\theta. \tag{14}$$

**Modeling momentum.** Momentum is a common extension to SGD that uses an exponentially moving average of gradients to update parameters rather than a single gradient evaluation Rumelhart et al. (1986). The method introduces two additional hyperparameters, a damping coefficient $\alpha$ and a momentum coefficient $\beta$, and applies the two step update equation, $\theta^{(n+1)} = \theta^{(n)} - \eta v^{(n+1)}$ where $v^{(n+1)} = \beta v^{(n)} + (1-\alpha)g(\theta^{(n)})$. When $\alpha = \beta = 0$, we regain classic gradient descent. In general, $\alpha$ effectively reduces the learning rate and $\beta$ controls how past gradients are used in future updates resulting in a form of "inertia" accelerating and smoothing the descent trajectory. Rearranging the two-step update equation, we find that gradient descent with momentum is a first-order discretization with step size $\eta(1-\alpha)$ of the ODE[1],

$$(1-\beta)\frac{d\theta}{dt} = -g(\theta). \tag{15}$$

**Modeling stochasticity.** Stochastic gradients $\hat{g}_{\mathcal{B}}(\theta)$ arise when we consider a batch $\mathcal{B}$ of size $S$ drawn uniformly from the indices $\{1, \dots, N\}$ forming the unbiased gradient estimate $\hat{g}_{\mathcal{B}}(\theta) = \frac{1}{S}\sum_{i \in \mathcal{B}} \nabla \ell(\theta, x_i)$. When the batch size is much smaller than the size of the dataset, $S \ll N$, then we can model the batch gradient as an average of $S$ i.i.d. samples from a noisy version of the true gradient $g(\theta)$. Using the central limit theorem, we assume $\hat{g}_{\mathcal{B}}(\theta) - g(\theta)$ is a Gaussian random variable with mean $\mu = 0$ and covariance matrix $\Sigma(\theta)$. However, because both the batch gradient and true gradient observe the same geometric properties introduced by symmetry, the noise has a special low-rank structure. As we showed in section 3, the gradient of the loss, regardless of the batch, is orthogonal to the generator vector field $\partial_\alpha \psi|_{\alpha=I}$ associated with a symmetry. This implies the stochastic noise must also observe the same property. In order for this relationship to hold for arbitrary noise, then $\Sigma(\theta)\partial_\alpha \psi|_{\alpha=I} = 0$. In other words, the differential symmetry inherent in neural network architectures projects the noise introduced by stochastic gradients onto low rank subspaces, leaving the gradient flow dynamics in these directions unchanged[2].

**Modeling discretization.** The effect of discretization when modeling continuous dynamics is a well studied problem in the numerical analysis of partial differential equations. One tool commonly used in this setting, is modified equation analysis Warming & Hyett (1974), which determines how to better model discrete steps with a continuous differential equation by introducing higher order spatial or temporal derivatives. We present two methods based on modified equation analysis, which modify gradient flow to account for the effect of discretization.

---

[1] A more detailed derivation for this ODE can be found in appendix C.

[2] Stochasticity and discretization can lead to non-trivial interactions requiring analysis using stochastic differential equation as explained in appendix F.

*Modified loss.* Gradient descent always moves in the direction of steepest descent on a loss function $\mathcal{L}$ at each step, however, due to the finite nature of the learning rate, it fails to remain on the continuous steepest descent path given by gradient flow. Li et al. (2017); Feng et al. (2019) and most recently Barrett & Dherin (2020), demonstrate that the gradient descent trajectory closely follows the steepest descent path of a modified loss function $\widetilde{\mathcal{L}}$. The divergence between these trajectories fundamentally depends on the learning rate $\eta$ and the curvature $H$. As derived in Barrett & Dherin (2020), and summarized in appendix D, this divergence is given by the gradient correction $-\frac{1}{2}Hg$, which is the gradient of the squared norm $-\frac{\eta}{4}|\nabla\mathcal{L}|^2$. Thus, the modified loss is $\widetilde{\mathcal{L}} = \mathcal{L} + \frac{\eta}{4}|\nabla\mathcal{L}|^2$ and the modified gradient flow ODE is

$$\frac{d\theta}{dt} = -g(\theta) - \frac{\eta}{2}H(\theta)g(\theta). \quad (16)$$

See Fig. 4 for an illustrative example of this method applied to a quadratic loss in $\mathbb{R}^2$.

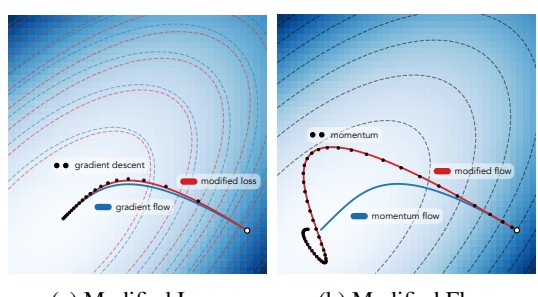

(a) Modified Loss  (b) Modified Flow

Figure 4: **Modeling discretization.** We visualize the trajectories of gradient descent and momentum (black dots), gradient flow with and without momentum (blue lines), and the modified dynamics (red lines) on the quadratic loss $\mathcal{L}(w) = w^\intercal \left[ \begin{smallmatrix} 2.5 & -1.5 \\ -1.5 & 2 \end{smallmatrix} \right] w$. On the left we visualize gradient dynamics using modified loss. On the right we visualize momentum dynamics using modified flow. In both settings the modified continuous dynamics visually track the discrete dynamics better than the original continuous dynamics. See appendix D for further details.

*Modified flow.* Rather than modifying gradient flow with higher order "spatial" derivatives of the loss function, here we introduce higher order temporal derivatives. We start by assuming the existence of a continuous trajectory $\theta(t)$ that weaves through the discrete steps taken by gradient descent and then identify the differential equation that generates the trajectory. Rearranging the update equation for gradient descent, $\theta_{t+1} = \theta_t - \eta g(\theta_t)$, and assuming $\theta(t) = \theta_t$ and $\theta(t+\eta) = \theta_{t+1}$, gives the equality $-g(\theta_t) = \frac{\theta(t+\eta)-\theta(t)}{\eta}$, which Taylor expanding the right side results in the differential equation $-g(\theta_t) = \frac{d\theta}{dt} + \frac{\eta}{2}\frac{d^2\theta}{dt^2} + O(\eta^2)$. Notice that in the limit as $\eta \to 0$ we regain gradient flow. For small $\eta$, we obtain a modified version of gradient flow with an additional second-order term,

$$\frac{d\theta}{dt} = -g(\theta) - \frac{\eta}{2}\frac{d^2\theta}{dt^2}. \quad (17)$$

This approach to modifying first-order differential equation with higher order temporal derivatives was applied by Kovachki & Stuart (2019) to construct a more realistic continuous model for momentum, as illustrated in Fig. 4.

## 6 COMBINING SYMMETRY AND MODIFIED GRADIENT FLOW TO DERIVE EXACT LEARNING DYNAMICS

As shown in section 4, each symmetry results in a conserved quantity under gradient flow. We now study how weight decay, momentum, stochastic gradients, and finite learning rates all interact to break these conservation laws. Remarkably, even when using a more realistic continuous model for stochastic gradient descent, as discussed in section 5, we can derive exact learning dynamics for the previously conserved quantities. To do this we (i) consider a realistic continuous model for SGD, (ii) project these learning dynamics onto the generator vector fields $\partial_\alpha\psi|_{\alpha=I}$ associated with each symmetry, (iii) harness the geometric constraints from section 3 to derive simplified ODEs, and (iv) solve these ODEs to obtain exact dynamics for the previously conserved quantities. For simplicity, we first consider the continuous model of SGD incorporating weight decay and modified loss, but not momentum and stochasticity[3]. In this setting, the exact dynamics, as fully derived in appendix E, for the parameter combinations tied to the symmetries are,

---

[3]See appendix F for a discussion on how to handle stochasticity in this setting using Itô calculus.

**Translation:** $\quad \langle \theta(t), \mathbb{1}_{\mathcal{A}} \rangle = e^{-\lambda t} \langle \theta(0), \mathbb{1}_{\mathcal{A}} \rangle$ $\hfill (18)$

**Scale:** $\quad |\theta_{\mathcal{A}}(t)|^2 = e^{-2\lambda t}|\theta_{\mathcal{A}}(0)|^2 + \eta \int_0^t e^{-2\lambda(t-\tau)} |g_{\mathcal{A}}|^2 \, d\tau$ $\hfill (19)$

**Rescale:** $\quad |\theta_{\mathcal{A}_1}(t)|^2 - |\theta_{\mathcal{A}_2}(t)|^2 =$ $\hfill (20)$

$$e^{-2\lambda t}(|\theta_{\mathcal{A}_1}(0)|^2 - |\theta_{\mathcal{A}_2}(0)|^2) + \eta \int_0^t e^{-2\lambda(t-\tau)} \left( \left|g_{\theta_{\mathcal{A}_1}}\right|^2 - \left|g_{\theta_{\mathcal{A}_2}}\right|^2 \right) d\tau$$

Notice how these equations are equivalent to the conservation laws derived in section 4 when $\eta = \lambda = 0$. Remarkably, even in typical hyperparameter settings (weight decay, stochastic batches, finite learning rates), these solutions match nearly perfectly with empirical results[4] from modern neural networks (VGG-16) trained on real-world datasets (Tiny ImageNet), as shown in Fig. 5. We will now discuss each equation individually.

**Translation dynamics.** For parameters with translation symmetry, equation 18 implies that the sum of these parameters ($\langle \theta_{\mathcal{A}}(t), \mathbb{1} \rangle$) decays exponentially to zero at a rate proportional to the weight decay. Equation 18 does not directly depend on the learning rate $\eta$ nor any information of the dataset or task. This is due to the lack of curvature in the gradient field for these parameters (as shown in Fig. 2). This implies that at initialization we can deterministically predict the trajectory for the parameter sum as simple exponential functions with a rate defined by the weight decay. The first row in Fig. 5 demonstrates this qualitatively, as all trajectories are smooth exponential functions that converge faster for increasing levels of weight decay.

**Scale dynamics.** For parameters with scale symmetry, equation 19 implies that the norm for these parameters ($|\theta_{\mathcal{A}}|^2$) is the sum of an exponentially decaying memory of the norm at initialization and an exponentially weighted integral of gradient norms accumulated through training. Compared to the translation dynamics, the scale dynamics do depend on the data through the gradient norms accumulated throughout training. Without weight decay $\lambda = 0$, the first term stays constant and the second term grows monotonically. With weight decay $\lambda > 0$, the first term decays monotonically to zero, while the second term can decay or grow, but always stays positive. The second row in Fig. 5 demonstrates these qualitative relationships. Without weight decay the norms increase monotonically as predicted and with weight decay the dynamics are non-monotonic and present more complex behavior. To better understand the forces driving these complex dynamics, we can examine the time derivative of equation 19,

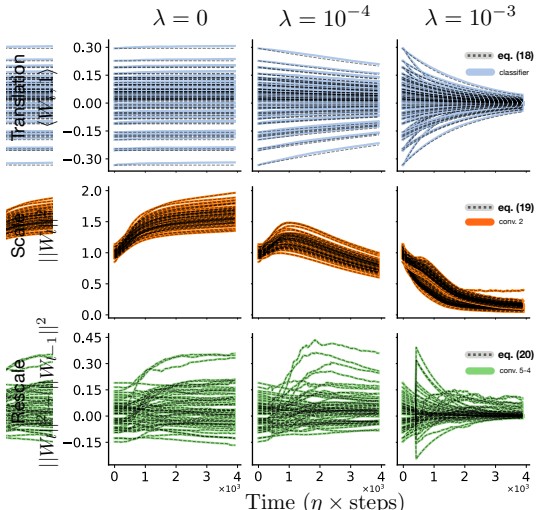

Figure 5: **Exact dynamics of VGG-16 on Tiny ImageNet.** We plot the column sum of the final linear layer (top row) and the difference between squared channel norms of the fifth and fourth convolutional layer (bottom row) of a VGG-16 model without batch normalization. We plot the squared channel norm of the second convolution layer (middle row) of a VGG-16 model with batch normalization. Both models are trained on Tiny ImageNet with SGD with learning rate $\eta = 0.1$, weight decay $\lambda$, batch size $S = 256$, for 100 epochs . Colored lines are empirical and black dashed lines are the theoretical predictions from equations (18), (19), and (20). See appendix J for more details on the experiments.

$$\frac{d}{dt}|\theta_{\mathcal{A}}(t)|^2 = -2\lambda|\theta_{\mathcal{A}}(t)|^2 + \eta \, |g_{\mathcal{A}}|^2 .$$ $\hfill (21)$

From this equation we see that there is a competition between a centripetal effect due to weight decay ($-2\lambda|\theta_{\mathcal{A}}(t)|^2$) and a centrifugal effect due to discretization ($\eta \, |g_{\mathcal{A}}|^2$). The centripetal effect

---

[4]See appendix J for more details on the experiments and how we compute the integrals terms in the exact solutions using stochastic gradients.

due to weight decay is a direct consequence of its regularizing influence, pulling the parameters towards the origin. The centrifugal effect due to discretization originates from the spherical geometry of the gradient field in parameter space – because scale symmetry implies the gradient is always orthogonal to the parameter itself, each discrete update with a finite learning rate effectively pushes the parameters away from the origin. At the stationary state of the dynamics, these forces will balance leading the dynamics of these parameters to be constrained to the surface of a high-dimensional sphere. In particular, at stationarity, then $\frac{d}{dt}|\theta(t)|^2 = 0$, which rearranging equation (21) gives the condition $\omega(t) \equiv \left| \frac{d\theta}{dt} \right| / |\theta| = \sqrt{\frac{2\lambda}{\eta}}$. Consistent with the results of Wan et al. (2020), this implies that at stationarity the angular speed $\omega(t)$ of the weights is constant and governed only by the learning rate $\eta$ and weight decay constant $\lambda$. When considering stochasticity explicitly, as explained in appendix F, then these dynamics also depend on the covariance of the gradient noise $\Sigma(\theta)$ and batch size $S$.

**Rescale dynamics.** For parameters with rescale symmetry, equation (20) is the sum of an exponentially decaying memory of the difference in norms at initialization and an exponentially weighted integral of difference in gradient norms accumulated through training. Similar to the scale dynamics, the rescale dynamics do depend on the data through the gradient norms, however unlike the scale dynamics we have no guarantee that the integral term is always positive. This leads to quite sophisticated, complex dynamics, consistent with the third row in Fig. 5. Despite the complexity, our theory, nevertheless, quantitatively matches the empirics. The only apparent pattern from the empirics is that for large enough weight decay, the regularization dominates any complexity introduced by the gradient norms and the difference in parameter norms decays exponentially to zero.

**Harmonic oscillation with momentum.** We will now consider a continuous model of SGD *with* momentum. As discussed in appendix G, we consider the continuous model incorporating weight decay, momentum, stochasticity, and modified flow. Under this model, the solutions we obtain take the form of driven harmonic oscillators where the driving force is given by the gradient norms, the friction is defined by the momentum constant, the spring coefficient is defined by the regularization rate, and the mass is defined by the the learning rate and momentum constant. For most standard hyperparameter choices, these solutions are in the overdamped setting and align well with the first-order solutions for SGD without momentum up to a time rescaling, as shown in the left and middle panel of Fig. 6. However, for large values of beta we can push the solution into the underdamped regime where we would expect harmonic oscillation and indeed, we can empirically verify our predictions, even at scale for VGG-16 trained on Tiny ImageNet, as in right panel of Fig. 6.

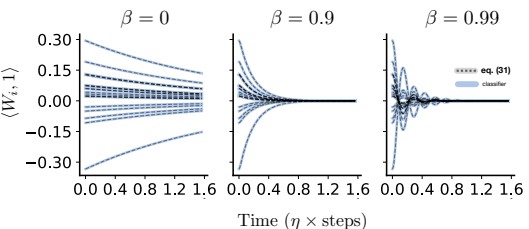

Figure 6: **Momentum leads to harmonic oscillation.** We plot the column sum of the final linear layer of a VGG-16 model (without batch normalization) trained on Tiny ImageNet with SGD with learning rate $\eta = 0.1$, weight decay $\lambda = 5 \times 10^{-3}$, batch size $S = 256$ and momentum $\beta \in \{0, 0.9, 0.99\}$. Colored lines are empirical and black dashed lines are the theoretical predictions from equations (34).

## 7 CONCLUSION

Despite being the central guiding principle in the exploration of the physical world Anderson (1972); Gross (1996), symmetry has been underutilized in understanding the mechanics of neural networks. In this paper, we constructed a unifying theoretical framework harnessing the geometric properties of symmetry and more realistic continuous equations for learning dynamics that model weight decay, momentum, stochasticity, and discretization. We use this framework to derive *exact* dynamics for meaningful combinations of parameters, which we experimentally verified on large scale neural networks and datasets. For example, in the case of a VGG-16 model with batch normalization trained on Tiny-ImageNet (one of the model/dataset combinations we considered in section 6) there are $12,751$ distinct parameter combinations whose dynamics we can analytically describe. Overall, this work provides a first step towards understanding the mechanics of learning in neural networks without unrealistic simplifying assumptions.

ACKNOWLEDGEMENTS

We thank Daniel Bear, Lauren Gillespie, Kyogo Kawaguchi, Brett Larsen, Eshed Margalit, Alain Studer, Sho Sugiura, Umberto Maria Tomasini, and Atsushi Yamamura for helpful discussions. This work was funded in part by the IBM-Watson AI Lab. D.K. thanks the Stanford Data Science Scholars program for support. J.S. thanks the Mexican National Council of Science and Technology (CONA-CYT) for support. S.G. thanks the James S. McDonnell and Simons Foundations, NTT Research, and an NSF CAREER Award for support. D.L.K.Y thanks the McDonnell Foundation (Understanding Human Cognition Award Grant No. 220020469), the Simons Foundation (Collaboration on the Global Brain Grant No. 543061), the Sloan Foundation (Fellowship FG-2018-10963), the National Science Foundation (RI 1703161 and CAREER Award 1844724), and the DARPA Machine Common Sense program for support and the NVIDIA Corporation for hardware donations.

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

# A    SYMMETRY AND GEOMETRY

Here we derive in detail the geometric properties of the loss landscape introduced by symmetry, as discussed in section 3. Consider a function $f(\theta)$ where $\theta \in \mathbb{R}^m$. This function possesses a symmetry if it is invariant under the action $\psi$ of a group $G$ on the parameter vector $\theta$, i.e., if $\theta \mapsto \psi(\theta, \alpha)$ where $\alpha \in G$, then $F(\theta, \alpha) = f(\psi(\theta, \alpha)) = f(\theta)$ for all $(\theta, \alpha)$. Symmetry enforces a geometric relationship between the gradient $\nabla F$ and Hessian $\mathbf{H}F$ of the composition $F(\theta, \alpha)$ with the gradient $\nabla f$ and Hessian $\mathbf{H}f$ of the original function $f(\theta)$. This relationship can be described by five constraints on $\nabla f$ and $\mathbf{H}f$. Considering these general formulae when $f(\theta) = \mathcal{L}(\theta)$, the training loss of a neural network, yields fifteen distinct equations describing the geometrical relationships between architectural symmetries and the loss landscapes, some of which have been identified individually in existing literature (Table 1).

| | | Translation | Scale | Rescale |
|---|---|---|---|---|
| $\nabla F$ | $\partial_\theta F$ | — | 1, 2, 5 | 6 |
| | $\partial_\alpha F$ | — | 3, 4 | 7,8,9 |
| $\mathbf{H}F$ | $\partial_\theta^2 F$ | — | 3, 5 | — |
| | $\partial_\theta \partial_\alpha F$ | — | — | — |
| | $\partial_\alpha^2 F$ | — | — | 9 |

Table 1: **Unifying existing literature through symmetry.** Here we provide references to existing literature describing geometric properties of either the gradient or Hessian introduced by a network's architecture. All of these properties can be unified as consequences of either a translation, scale, or rescale symmetry in the training loss.

1. Ioffe & Szegedy (2015) has motivated the effectiveness of Batch Normalization by its scale invariant property. In particular, they have noted that Batch Normalization will stabilize back-propagation because "The scale does not affect the layer Jacobian nor, consequently, the gradient propagation. Moreover, larger weights lead to smaller gradients, and Batch Normalization will stabilize the parameter growth."

2. Van Laarhoven (2017) has then shown that the role of $L_2$ regularization when combined with batch Ioffe & Szegedy (2015), weight Salimans & Kingma (2016) or layer Ba et al. (2016) normalization is not to regularize the function, but to effectively control the learning rate.

3. Zhang et al. (2018) has thoroughly studied mechanisms of weight decay regularization and derived various geometric properties of loss landscapes along the way. Arora et al. (2018c) has theoretically analyzed the automatic tuning property of learning rate in networks with Batch Normalization.

4. Li et al. (2020) studied the interaction of weight decay and batch normalization in the setting of stochastic gradients.

5. Neyshabur et al. (2015) has identified that SGD is not rescale equivariant even when network outputs are rescale invariant. This is a problem because gradient descent performs very poorly on unbalanced networks due to the lack of equivariance. Motivated by the issue, they have introduced a new optimizer, Path-SGD, that is rescale equivariant.

6. Arora et al. (2018b) proved that the weights of linear artificial neural networks satisfy strong balancedness property.

7. Du et al. (2018a) studied implicit regularization in networks with homogeneous activation functions. To do that, they showed conservation law of parameters with rescale invariance.

8. Liang et al. (2019) have proposed a new capacity measure to study generalization that respects rescale invariance of networks. Along the way, they showed geometric properties of gradients and Hessians for networks with rescale invariance. However, their results were restricted to a layer without biases.

9. Tanaka et al. (2020) have proved gradient properties of parameters with rescale invariance at neuron level including biases.

## A.1 GRADIENT GEOMETRY

If a function $f$ posses a symmetry, then there exists a geometric constraint on the relationship between the gradients $\nabla F$ and $\nabla f$ at all $(\theta, \alpha)$,

$$\nabla F = \begin{pmatrix} \partial_\theta F \\ \partial_\alpha F \end{pmatrix} = \begin{pmatrix} \partial_\psi F \partial_\theta \psi \\ \partial_\psi F \partial_\alpha \psi \end{pmatrix} = \begin{pmatrix} \nabla f \\ 0 \end{pmatrix}.$$

The top element of the gradient relationship, $\partial_\theta F$, evaluated at any $(\theta, \alpha)$, yields the property

$$\partial_\theta \psi \, \nabla f|_{\psi(\theta,\alpha)} = \nabla f|_\theta, \tag{22}$$

which describes how the symmetry transformation affects the function's gradients despite leaving the output unchanged. The bottom element of the gradient relationship, $\partial_\alpha F$, evaluated at the identity element of $G$ yields the property

$$\langle \nabla f, \partial_\alpha \psi|_{\alpha=I} \rangle = 0, \tag{23}$$

which implies the gradient $\nabla f$ is perpendicular to the vector field $\partial_\alpha \psi|_{\alpha=I}$ that generates the symmetry, for all $\theta$. In the specific setting when $f(\theta) = \mathcal{L}(\theta)$, the training loss of a neural network, these gradient properties are summarized in Table 2 for the translation, scale, and rescale symmetries described in section 3.

| | Translation | Scale | Rescale |
|---|---|---|---|
| $g(\theta) =$ | $g(\psi(\theta,\alpha))$ | $\text{diag}(\alpha_\mathcal{A})g(\psi(\theta,\alpha))$ | $\text{diag}(\alpha_{\mathcal{A}_1} \odot \alpha_{\mathcal{A}_2}^{-1})g(\psi(\theta,\alpha))$ |
| $g(\theta) \perp$ | $\mathbb{1}_\mathcal{A}$ | $\theta_\mathcal{A}$ | $\theta_{\mathcal{A}_1} - \theta_{\mathcal{A}_2}$ |

Table 2: **Geometric properties of the gradient.** The gradients of a neural network with either translation, scale or rescale symmetry observe certain geometric properties no matter the dataset or step in training.

Notice that the first row of Table 2 implies that symmetry transformations affect learning dynamics governed by gradient descent for scale and rescale symmetries, while it does not for translation symmetry. These observations are in agreement with Van Laarhoven (2017) who has shown that effective learning rate is inversely proportional to the norm of parameters immediately preceding the batch normalization layers and Neyshabur et al. (2015) who have noticed that SGD is not invariant to the rescale symmetry that the network output respects and proposed Path-SGD to fix the discrepancy.

## A.2 HESSIAN GEOMETRY

If a function $f$ posses a symmetry, then there also exists a geometric constraint on the relationship between the Hessian matrices $\mathbf{H}F$ and $\mathbf{H}f$ at all $(\theta, \alpha)$,

$$\mathbf{H}F = \begin{pmatrix} \partial_\theta^2 F & \partial_\theta \partial_\alpha F \\ \partial_\alpha \partial_\theta F & \partial_\alpha^2 F \end{pmatrix}$$
$$= \begin{pmatrix} \partial_\psi F \partial_\theta^2 \psi + \partial_\psi^2 F (\partial_\theta \psi)^2 & \partial_\psi^2 F \partial_\theta \psi \partial_\alpha \psi + \partial_\psi F \partial_\theta \partial_\alpha \psi \\ \partial_\psi^2 F \partial_\theta \psi \partial_\alpha \psi + \partial_\psi F \partial_\theta \partial_\alpha \psi & (\partial_\alpha \psi)^\intercal \partial_\psi^2 F \partial_\alpha \psi + (\partial_\psi F)^\intercal \partial_\alpha^2 \psi \end{pmatrix} = \begin{pmatrix} \mathbf{H}f & 0 \\ 0 & 0 \end{pmatrix}.$$

The first diagonal element, $\partial_\theta^2 F$, evaluated at any $(\theta, \alpha)$, yields the property

$$\partial_\theta^2 \psi \, \nabla f|_{\psi(\theta,\alpha)} + (\partial_\theta \psi)^2 \, \mathbf{H}f|_{\psi(\theta,\alpha)} = \mathbf{H}f|_\theta, \tag{24}$$

which describes how the symmetry transformation affects the function's Hessian despite leaving the output unchanged. The off-diagonal elements, $\partial_\theta \partial_\alpha F = \partial_\alpha \partial_\theta F$, evaluated at the identity element of $G$ yields the property

$$\mathbf{H}f[\partial_\theta \psi|_{\alpha=I}]\partial_\alpha \psi|_{\alpha=I} + [\partial_\theta \partial_\alpha|_{\alpha=I}]\psi \nabla f = 0, \tag{25}$$

which implies the geometry of gradient and Hessian are connected through the action of the symmetry. Lastly, the second diagonal element, $\partial_\alpha^2 F$, represents an equality, evaluated at the identity element of $G$ yields the property

$$(\partial_\alpha \psi|_{\alpha=I})^\intercal \mathbf{H}f(\partial_\alpha \psi|_{\alpha=I}) + \langle \nabla f, \partial_\alpha^2 \psi|_{\alpha=I} \rangle = 0, \tag{26}$$

which combines the geometric relationships in equation 23 and equation 25. In the specific setting when $f(\theta) = \mathcal{L}(\theta)$, the training loss of a neural network, these Hessian properties are summarized in Table 2 for the translation, scale, and rescale symmetries described in section 3.

| | Translation | Scale | Rescale |
|---|---|---|---|
| $H(\theta) =$ | $H(\psi(\theta, \alpha))$ | $\mathrm{diag}(\alpha_{\mathcal{A}}^2) H(\psi(\theta, \alpha))$ | $\mathrm{diag}(\alpha_{\mathcal{A}_1}^2 \odot \alpha_{\mathcal{A}_2}^{-2}) H(\psi(\theta, \alpha))$ |
| $0 =$ | $H \mathbb{1}_{\mathcal{A}}$ | $H\theta_{\mathcal{A}} + g_{\mathcal{A}}$ | $H(\theta_{\mathcal{A}_1} - \theta_{\mathcal{A}_2}) + g_{\mathcal{A}_1} - g_{\mathcal{A}_2}$ |
| $0 =$ | $\mathbb{1}_{\mathcal{A}}^{\intercal} H \mathbb{1}_{\mathcal{A}}$ | $\theta_{\mathcal{A}}^{\intercal} H \theta_{\mathcal{A}}$ | $(\theta_{\mathcal{A}_1} - \theta_{\mathcal{A}_2})^{\intercal} H (\theta_{\mathcal{A}_1} - \theta_{\mathcal{A}_2}) + g_{\mathcal{A}_1}\theta_{\mathcal{A}_1} + g_{\mathcal{A}_2}\theta_{\mathcal{A}_2}$ |

Table 3: **Geometric properties of the Hessian.** The Hessian matrix of a neural network with either translation, scale or rescale symmetry observe certain geometric properties no matter the dataset or step in training.

The translation, scale, and rescale symmetries identified in section 3 is not an exhaustive list of the symmetries present in neural network architectures. For example, more general rescale symmetries can be defined by the group $\mathrm{GL}_1^+(\mathbb{R})$ and action $\theta \mapsto \alpha_{\mathcal{A}_1}^{k_1} \odot \alpha_{\mathcal{A}_2}^{-k_2} \odot \theta$, which occur in networks with quadratic activation functions. A stronger form of rescale symmetry also occurs for linear networks under the action of the group $GL_k^+(\mathbb{R})$ of $k \times k$ invertible matrices, as noticed previously by Arora et al. (2018a); Du et al. (2018a). Interestingly, some of the gradient and Hessian properties for scale symmetry can also be easily proven as consequences of Euler's Homogeneous Function Theorem when $k = 0$.

## B    CONSERVATION LAWS

Here we repeat Theorem 1 and provide a detailed derivation.

**Theorem.** *Symmetry and conservation laws in neural networks. Every differentiable symmetry* $\psi(\alpha, \theta)$ *of the loss that satisfies* $\langle \theta, [\partial_\alpha \partial_\theta \psi|_{\alpha=I}] g(\theta) \rangle = 0$ *has the corresponding conservation law* $\frac{d}{dt}\langle \theta, \partial_\alpha \psi|_{\alpha=I} \rangle = 0$ *through learning under gradient flow.*

*Proof.* Project the gradient flow learning dynamics, $\frac{d\theta}{dt} = -g(\theta)$, onto the vector field that generates the symmetry $\partial_\alpha \psi|_{\alpha=I}$, evaluated at the identity element,

$$\langle \frac{d\theta}{dt}, \partial_\alpha \psi|_{\alpha=I} \rangle = \langle -g(\theta), \partial_\alpha \psi|_{\alpha=I} \rangle = 0.$$

We can factor the left side of this equation as

$$\langle \frac{d\theta}{dt}, \partial_\alpha \psi|_{\alpha=I} \rangle = \frac{d}{dt}\langle \theta, \partial_\alpha \psi|_{\alpha=I} \rangle - \langle \theta, \frac{d}{dt}\partial_\alpha \psi|_{\alpha=I} \rangle$$

$$= \frac{d}{dt}\langle \theta, \partial_\alpha \psi|_{\alpha=I} \rangle - \langle \theta, \partial_\alpha \partial_\theta \psi|_{\alpha=I} \frac{d\theta}{dt} \rangle$$

$$= \frac{d}{dt}\langle \theta, \partial_\alpha \psi|_{\alpha=I} \rangle + \langle \theta, \partial_\alpha \partial_\theta \psi|_{\alpha=I} g(\theta) \rangle$$

By assumption, $\langle \theta, [\partial_\alpha \partial_\theta \psi|_{\alpha=I}] g(\theta) \rangle = 0$, implying $\frac{d}{dt}\langle \theta, \partial_\alpha \psi|_{\alpha=I} \rangle = 0$.     $\square$

The condition $\langle \theta, [\partial_\alpha \partial_\theta \psi|_{\alpha=I}] g(\theta) \rangle = 0$ holds for the translation, scale, and rescale symmetries we consider in section 3. For translation symmetry, $\partial_\alpha \partial_\theta \psi|_{\alpha=I} = 0$. For scale symmetry, $\partial_\alpha \partial_\theta \psi|_{\alpha=I} = I$ and $\langle \theta_{\mathcal{A}}, g(\theta_{\mathcal{A}}) \rangle = 0$. For rescale symmetry, $\partial_\alpha \partial_\theta \psi|_{\alpha=I} = \begin{bmatrix} I & 0 \\ 0 & -I \end{bmatrix}$ and $\langle \theta_{\mathcal{A}_1}, g(\theta_{\mathcal{A}_1}) \rangle - \langle \theta_{\mathcal{A}_2}, g(\theta_{\mathcal{A}_2}) \rangle = 0$.

## C   LIMITING DIFFERENTIAL EQUATIONS FOR LEARNING RULES

Here we identify ordinary differential equations whose first-order discretization give rise to the gradient descent and classical momentum algorithms. These differential equations can be understood as the limiting dynamics for their respective discrete algorithms as the learning rate $\eta \to 0$.

### C.1   GRADIENT DESCENT

Gradient descent with learning rate $\eta$ is given by the update equation

$$\theta_{k+1} = \theta_k - \eta g(\theta_k),$$

and initial condition $\theta_0$. Rearranging the difference between consecutive updates gives

$$\frac{\theta_{k+1} - \theta_k}{\eta} = -g(\theta_k).$$

This is a discretization with step size $\eta$ of the first order ODE

$$\frac{d}{dt}\theta = -g(\theta),$$

where we used the forward Euler discretization $\frac{d}{dt}\theta_k = \frac{\theta_{k+1} - \theta_k}{\eta}$. This ODE is commonly referred to as *gradient flow*.

### C.2   CLASSICAL MOMENTUM

Classical momentum with learning rate $\eta$, damping coefficient $\alpha$, and momentum parameter $\beta$, is given by the update equation[5]

$$v_{k+1} = \beta v_k + (1 - \alpha)g(\theta_k),$$
$$\theta_{k+1} = \theta_k - \eta v_{k+1},$$

and initial conditions $v_0 = 0$ and some $\theta_0$. The difference between consecutive updates is

$$
\begin{aligned}
\theta_{k+1} - \theta_k &= -\eta v_{k+1} \\
&= -\eta \beta v_k - \eta(1 - \alpha)g(\theta_k) \\
&= \beta\left(\theta_k - \theta_{k-1}\right) - \eta(1 - \alpha)g(\theta_k).
\end{aligned}
$$

Rearranging this equation we get

$$\frac{\theta_{k+1} - \theta_k}{\eta(1 - \alpha)} - \beta\frac{\theta_k - \theta_{k-1}}{\eta(1 - \alpha)} = -g(\theta_k).$$

This is a discretization with step size $\eta(1 - \alpha)$ of the first order ODE

$$(1 - \beta)\frac{d}{dt}\theta = -g(\theta),$$

where we used the forward Euler discretization $\frac{d}{dt}\theta_k = \frac{\theta_{k+1} - \theta_k}{\eta(1-\alpha)}$ and backward Euler discretization $\frac{d}{dt}\theta_k = \frac{\theta_k - \theta_{k-1}}{\eta(1-\alpha)}$. We will refer to this equation as *momentum flow*. A more detailed derivation for this ODE under Nesterov variants of classical momentum can be found in Su et al. (2014) and Kovachki & Stuart (2019).

---

[5]The default PyTorch implementation does not perform damping on the gradient in the first momentum buffer $v_1$.

# D MODIFIED EQUATION ANALYSIS

Gradient descent always moves in the direction of steepest descent on a loss function, however, due to the finite nature of the learning rate, it fails to remain on the continuous steepest descent path. The divergence between the discrete and continuous trajectories fundamentally depends on the learning rate and the curvature of the loss. It is thus natural to assume there exists a more realistic continuous model for SGD that incorporates both these terms in a non-trivial way. *How can we better model the discrete dynamics of gradient descent with a continuous differential equation?*

**Intuition from finite difference methods.** To answer this question we will take inspiration from tools developed for finite difference methods. Finite difference methods are a class of numerical techniques for approximating derivatives in the analysis of partial differential equations (PDE). These approximations are applied iteratively to construct numerical solutions to a PDE given some initial conditions. However, this discretization process introduces numerical artifacts, which can lead to a significant difference between the numerical solution and the true solution for the PDE. Modified equation analysis is a method for understanding this difference by modeling the numerical artifacts as higher-order spatial or temporal derivatives modifying the original PDE. This approach can be used to construct modified continuous dynamics that better approximate discrete dynamics, as illustrated in Fig. 7.

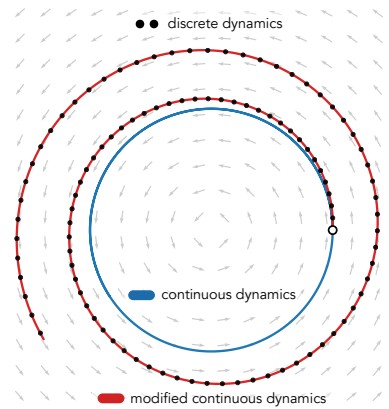

Figure 7: **Circular motion.** Consider the vector field $f(x) = \begin{bmatrix} 0 & -1 \\ 1 & 0 \end{bmatrix} x$ and the discrete dynamics $x_{t+1} = x_t + \eta f(x_t)$ (black dots), the continuous dynamics $\dot{x} = f(x)$ (blue line), and the modified continuous dynamics $\dot{x} = f(x) + \frac{\eta}{2} x$. We visualize the trajectories given by these dynamics using the initial condition $x_0 = \begin{bmatrix} 1 & 0 \end{bmatrix}^\mathsf{T}$ (white circle) and a step size $\eta = 0.1$. As we can see, the modified continuous trajectory better matches the discrete trajectory.

## D.1 MODIFIED LOSS

Taking inspiration from modified equation analysis, Li et al. (2017); Feng et al. (2019) and most recently Barrett & Dherin (2020), demonstrate that the trajectory given by gradient descent closely follows the steepest descent path of a modified loss function $\widetilde{\mathcal{L}}$, rather than the original loss $\mathcal{L}$. As explained in Barrett & Dherin (2020), assume there exists a modified vector field with corrections $g_i$ in powers of the learning rate to the original vector field $g$ that the discrete dynamics follow. In other words, rather than considering the dynamics given by gradient flow, $\frac{d}{dt}\theta = -g(\theta)$, we consider the modified differential equation,

$$\frac{d}{dt}\theta = -g(\theta) + \eta g_1(\theta) + \eta^2 g_2(\theta) + \ldots.$$

Truncating the modified vector field up to the order $\eta$ and using backward error analysis we can derive that the first-order correction $g_1 = -\frac{1}{2}Hg$, which is the gradient of the squared norm $-\frac{\eta}{4}|\nabla\mathcal{L}|^2$. Thus, the truncated modified differential equation is simply gradient flow on a modified loss $\widetilde{\mathcal{L}} = \mathcal{L} + \frac{\eta}{4}|\nabla\mathcal{L}|^2$.

**Convex quadratic loss.** To illustrate modified loss, we will consider the trajectories of gradient descent $w_t$, gradient flow $\hat{w}(t)$, and modified gradient flow $\tilde{w}(t)$ on the convex quadratic loss $\mathcal{L}(w) = \frac{1}{2}w^\mathsf{T}Aw$, where $A \succ 0$ is some positive definite matrix, as shown in Fig. 4. For a finite learning rate $\eta$ and initial condition $w_0$, gradient descent is given by the update formula $w_{t+1} = w_t - \eta A w_t$. Gradient flow is defined as $\frac{d}{dt}\hat{w} = -A\hat{w}$, a linear first-order differential equation. For the initial condition $w_0$, the resulting initial value problem can be solved exactly giving the gradient flow trajectory $\hat{w}(t) = S^{-1}e^{-\Lambda t}Sw_0$, where $A = S^{-1}\Lambda S$ is the diagonalization of the curvacture matrix. For learning rate $\eta$, the modified loss is $\widetilde{\mathcal{L}}(w) = \frac{1}{2}w^\mathsf{T}Aw + \frac{\eta}{4}w^\mathsf{T}A^2w$ and modified gradient flow is defined as $\frac{d}{dt}\tilde{w} = -A\tilde{w} - \frac{\eta}{2}A^2\tilde{w}$. For the initial condition $w_0$, the resulting initial value problem can be solved exactly giving the modified gradient flow trajectory $\tilde{w}(t) = S^{-1}e^{-\left(\Lambda + \frac{\eta}{2}\Lambda^2\right)t}Sw_0$.

## D.2 MODIFIED FLOW

Rather than modify gradient flow with higher order "spatial" derivatives of the loss function, here we introduce higher order temporal derivatives. We start by assuming the existence of a continuous trajectory $\theta(t)$ that weaves through the discrete steps taken by SGD and then identify the differential equation that generates the trajectory. Rearranging the update equation for SGD, $\theta_{t+1} = \theta_t - \eta g_{\mathcal{B}}(\theta_t)$, and assuming $\theta(t) = \theta_t$ and $\theta(t + \eta) = \theta_{t+1}$, gives the equality $-g_{\mathcal{B}}(\theta_t) = \frac{\theta(t+\eta)-\theta(t)}{\eta}$, which Taylor expanding the right side results in the differential equation $-g_{\mathcal{B}}(\theta_t) = \frac{d\theta}{dt} + \frac{\eta}{2}\frac{d^2\theta}{dt^2} + O(\eta^2)$. Notice that in the limit as $\eta \to 0$ we regain gradient flow. For small $\eta$, $\eta \ll 1$, we obtain a modified version of gradient flow with an additional second-order term. This approach of modifying first-order differential equations with higher order temporal derivatives was applied by Kovachki & Stuart (2019) to modify momentum flow, capturing the harmonic motion of momentum.

**Convex quadratic loss.** To illustrate modified flow, we will consider the trajectories of momentum $w_t$, momentum flow $\hat{w}(t)$, and modified momentum flow $\tilde{w}(t)$ on the convex quadratic loss $\mathcal{L}(w) = \frac{1}{2}w^\intercal A w$, where $A \succ 0$ is some positive definite matrix, as shown in Fig. 4. For a finite learning rate $\eta$, dampening $\alpha = 0$, momentum constant $\beta$, and initial conditions $v_0 = 0, w_0$, then momentum is given by the pair of recursive update equations $v_{t+1} = \beta v_t + A w_t$ and $w_{t+1} = w_t - \eta v_{t+1}$. Momentum flow is defined as $(1 - \beta)\frac{d}{dt}\hat{w} = -A\hat{w}$, a linear first-order differential equation. For a given initialization $w_0$, the resulting initial value problem can be solved exactly as in the case of gradient flow. Modified momentum flow is defined as $\frac{\eta}{2}(1+\beta)\frac{d^2}{dt^2}\tilde{w} + (1-\beta)\frac{d}{dt}\tilde{w} = -A\tilde{w}$, a linear second-order differential equation. For a given initialization $w_0$ and the assumed initial condition $\frac{d}{dt}\tilde{w}(0) = 0$, then the resulting initial value problem can be solved exactly as a system of damped harmonic oscillators.

## E   DERIVING THE EXACT LEARNING DYNAMICS OF SGD

We consider a continuous model of SGD without momentum incorporating weight decay (equation 14) and modified loss (equation 16), such that

$$\widetilde{\mathcal{L}} = \mathcal{L}_\lambda + \frac{\eta}{4}|\nabla \mathcal{L}_\lambda|^2$$

where $\mathcal{L}_\lambda = \mathcal{L} + \frac{\lambda}{2}|\theta|^2$ is the regularized loss. The gradient of the modified loss is,

$$\nabla \widetilde{\mathcal{L}} = \nabla \mathcal{L}_\lambda + \frac{\eta}{2}H_\lambda \nabla \mathcal{L}_\lambda = \left(1 + \frac{\eta\lambda}{2}\right)g + \left(\lambda + \frac{\eta\lambda^2}{2}\right)\theta + \frac{\eta}{2}\left(Hg + \lambda H\theta\right),$$

where we used $\nabla \mathcal{L}_\lambda = g + \lambda\theta$ and $H_\lambda = H + \lambda I$. Thus, the equation of learning we consider is

$$\frac{d\theta}{dt} = -\nabla \widetilde{\mathcal{L}}(\theta) = -\left(1 + \frac{\eta\lambda}{2}\right)g - \left(\lambda + \frac{\eta\lambda^2}{2}\right)\theta - \frac{\eta}{2}\left(Hg + \lambda H\theta\right). \qquad (27)$$

To incorporate the effect of stochasticity (equation 31) in this equation of learning we could replace the full-batch gradient $g$ and Hessian $H$ with their stochastic batch counterparts $\hat{g}_{\mathcal{B}}$ and $\hat{H}_{\mathcal{B}}$ respectively. However, careful treatment of these terms using stochastic calculus is needed when integrating the resulting stochastic differential equation, which we discuss at the end of this section.

**Translation dynamics.** In the case of parameters with translation symmetry, the effect of discretization essentially leaves the dynamics for the constant of learning unchanged. Combining the geometric properties of gradient ($\langle g, \mathbb{1}_{\mathcal{A}} \rangle = 0$) and Hessian ($H\mathbb{1}_{\mathcal{A}} = 0$) introduced by translation symmetry with the equation of learning (equation 27) gives the differential equation,

$$\left\langle \frac{d\theta}{dt} + \nabla \widetilde{\mathcal{L}}, \mathbb{1}_{\mathcal{A}} \right\rangle = \left(\lambda + \frac{d}{dt}\right)\langle \theta, \mathbb{1}_{\mathcal{A}} \rangle = 0, \qquad (28)$$

where we used the simplification,

$$\langle \nabla \widetilde{\mathcal{L}}, \mathbb{1}_{\mathcal{A}} \rangle = \left(1 + \frac{\eta\lambda}{2}\right)\overbrace{\langle g, \mathbb{1}_{\mathcal{A}} \rangle}^{\langle g, \mathbb{1}_{\mathcal{A}} \rangle = 0} + \left(\lambda + \frac{\eta\lambda^2}{2}\right)\langle \theta, \mathbb{1}_{\mathcal{A}} \rangle + \frac{\eta}{2}\left(\overbrace{\langle Hg, \mathbb{1}_{\mathcal{A}} \rangle}^{H\mathbb{1}_{\mathcal{A}}=0} + \lambda \overbrace{\langle H\theta, \mathbb{1}_{\mathcal{A}} \rangle}^{H\mathbb{1}_{\mathcal{A}}=0}\right)$$

$$= \left(\lambda + \frac{\eta\lambda^2}{2}\right)\langle \theta, \mathbb{1}_{\mathcal{A}} \rangle,$$

and ignored the $O(\eta\lambda^2)$ term as we set the weight decay constant $\lambda$ to be as small as the learning rate $\eta$ in practice. The solution to this differential equation is

$$\langle \theta(t), \mathbb{1}_{\mathcal{A}} \rangle = e^{-\lambda t} \langle \theta(0), \mathbb{1}_{\mathcal{A}} \rangle.$$

**Scale dynamics.** In the case of parameters with scale symmetry, the effect of discretization does distort the original geometry. A finite learning rate leads to a centrifugal force that monotonically increases the previously conserved quantity $(|\theta|^2)$, while weight decay acts as a centripetal force decreasing the quantity. Combining the geometric constraints on the gradient $(\langle g, \theta_{\mathcal{A}} \rangle = 0)$ and Hessian $(H\theta_{\mathcal{A}} = -g_{\mathcal{A}})$ introduced by scale symmetry with the equation of learning (equation 27) gives the following differential equation,

$$\left\langle \frac{d\theta}{dt} + \nabla\widetilde{\mathcal{L}}, \theta_{\mathcal{A}} \right\rangle = \left( \lambda + \frac{1}{2}\frac{d}{dt} \right) |\theta_{\mathcal{A}}|^2 - \frac{\eta}{2}|g_{\mathcal{A}}|^2 = 0, \tag{29}$$

where we used the simplifications $\langle \frac{d\theta}{dt}, \theta_{\mathcal{A}} \rangle = \frac{1}{2}\frac{d}{dt}|\theta_{\mathcal{A}}|^2$,

$$\langle \nabla\widetilde{\mathcal{L}}, \theta_{\mathcal{A}} \rangle = \left( 1 + \frac{\eta\lambda}{2} \right) \overbrace{\langle g, \theta_{\mathcal{A}} \rangle}^{\langle g, \theta_{\mathcal{A}} \rangle = 0} + \left( \lambda + \frac{\eta\lambda^2}{2} \right) \langle \theta, \theta_{\mathcal{A}} \rangle + \frac{\eta}{2}( \overbrace{\langle Hg, \theta_{\mathcal{A}} \rangle}^{\langle g, H\theta_{\mathcal{A}} \rangle = -|g^2|} + \lambda \overbrace{\langle H\theta, \theta_{\mathcal{A}} \rangle}^{-\langle g, \theta_{\mathcal{A}} \rangle = 0} )$$

$$= \left( \lambda + \frac{\eta\lambda^2}{2} \right) |\theta_{\mathcal{A}}|^2 - \frac{\eta}{2}|g_{\mathcal{A}}|^2,$$

and ignored the $O(\eta\lambda^2)$ term. The solution to this differential equation is

$$|\theta_{\mathcal{A}}(t)|^2 = e^{-2\lambda t}|\theta_{\mathcal{A}}(0)|^2 + \eta \int_0^t e^{-2\lambda(t-\tau)}|g_{\mathcal{A}}|^2 \, d\tau.$$

**Rescale dynamics.** In the case of parameters with rescale symmetry, the effect of discretization also distorts the original geometry. However, unlike in the sphere, the force originating from discretization can both increase or decrease the previously conserved quantity $(|\theta_{\mathcal{A}_1}(t)|^2 - |\theta_{\mathcal{A}_2}(t)|^2)$. Combining the geometric properties of gradient $(\langle g, \theta_{\mathcal{A}_1} \rangle - \langle g, \theta_{\mathcal{A}_2} \rangle = 0)$ and Hessian $(H\theta_{\mathcal{A}_1} - H\theta_{\mathcal{A}_2} + g_{\mathcal{A}_1} - g_{\mathcal{A}_2} = 0)$ introduced by rescale symmetry with the equation of learning (equation 27) gives the following differential equation,

$$\left\langle \frac{d\theta}{dt} + \nabla\widetilde{\mathcal{L}}, \theta_{\mathcal{A}_1} \right\rangle - \left\langle \frac{d\theta}{dt} + \nabla\widetilde{\mathcal{L}}, \theta_{\mathcal{A}_2} \right\rangle = \left( \lambda + \frac{1}{2}\frac{d}{dt} \right) (|\theta_{\mathcal{A}_1}(t)|^2 - |\theta_{\mathcal{A}_2}(t)|^2) - \frac{\eta}{2} \left( |g_{\mathcal{A}_1}|^2 - |g_{\mathcal{A}_2}|^2 \right) = 0, \tag{30}$$

where we used the simplification,

$$\langle \nabla_\theta \widetilde{\mathcal{L}}, \theta_{\mathcal{A}_1} \rangle - \langle \nabla\widetilde{\mathcal{L}}, \theta_{\mathcal{A}_2} \rangle = \left( 1 + \frac{\eta\lambda}{2} \right) \overbrace{(\langle g, \theta_{\mathcal{A}_1} \rangle - \langle g, \theta_{\mathcal{A}_2} \rangle)}^{\langle g, \theta_{\mathcal{A}_1} \rangle - \langle g, \theta_{\mathcal{A}_2} \rangle = 0} + \left( \lambda + \frac{\eta\lambda^2}{2} \right) (|\theta_{\mathcal{A}_1}|^2 - |\theta_{\mathcal{A}_2}|^2)$$

$$+ \frac{\eta}{2} \left( \overbrace{\langle g, H\theta_{\mathcal{A}_1} - H\theta_{\mathcal{A}_2} \rangle}^{\langle g, -g_{\mathcal{A}_1} + g_{\mathcal{A}_2} \rangle = -|g_{\mathcal{A}_1}|^2 + |g_{\mathcal{A}_2}|^2} + \lambda \overbrace{\langle \theta, H\theta_{\mathcal{A}_1} - H\theta_{\mathcal{A}_2} \rangle}^{-\langle g, \theta_{\mathcal{A}_1} \rangle + \langle g, \theta_{\mathcal{A}_2} \rangle = 0} \right)$$

$$= \left( \lambda + \frac{\eta\lambda^2}{2} \right) (|\theta_{\mathcal{A}_1}|^2 - |\theta_{\mathcal{A}_2}|^2) - \frac{\eta}{2}(|g_{\mathcal{A}_1}|^2 - |g_{\mathcal{A}_2}|^2),$$

and ignored the $O(\eta\lambda^2)$ term. This is the same differential equation as in equation (29), just with a different forcing term. Thus, the solution to this differential equation is

$$|\theta_{\mathcal{A}_1}(t)|^2 - |\theta_{\mathcal{A}_2}(t)|^2 = e^{-2\lambda t}(|\theta_{\mathcal{A}_1}(0)|^2 - |\theta_{\mathcal{A}_2}(0)|^2) + \eta \int_0^t e^{-2\lambda(t-\tau)} \left( |g_{\mathcal{A}_1}|^2 - |g_{\mathcal{A}_2}|^2 \right) d\tau.$$

## F  MODELING STOCHASTICITY

Stochastic gradients $\hat{g}_{\mathcal{B}}(\theta)$ arise when we consider a batch $\mathcal{B}$ of size $S$ drawn uniformly from the indices $\{1, \ldots, N\}$ forming the unbiased gradient estimate $\hat{g}_{\mathcal{B}}(\theta) = \frac{1}{S} \sum_{i \in \mathcal{B}} \nabla \ell(\theta, x_i)$. When the batch size is much smaller than the size of the dataset, $S \ll N$, then we can model the batch gradient as an average of $S$ i.i.d. samples from a noisy version of the true gradient $g(\theta)$. Using the central limit theorem, we assume $\hat{g}_{\mathcal{B}}(\theta) - g(\theta)$ is a Gaussian random variable with mean $\mu = 0$ and covariance matrix $\Sigma(\theta) = \frac{1}{S} G(\theta) G(\theta)^\intercal$. Under this assumption, the stochastic gradient update can be written as $\theta^{(n+1)} = \theta^{(n)} - \eta g(\theta^{(n)}) + \frac{\eta}{\sqrt{S}} G(\theta) \xi$, where $\xi$ is a standard normal random variable. This update is an Euler-Maruyama discretization with step size $\eta$ of the stochastic differential equation

$$d\theta = -g(\theta)dt + \sqrt{\frac{\eta}{S}} G(\theta) dW_t, \tag{31}$$

where $W_t$ is a standard Wiener process. Equation (31) has been derived as a model for SGD in many previous works Mandt et al. (2015). In order to simplify the analysis, many of these works have then made additional assumptions on the covariance matrix $\Sigma(\theta) = G(\theta)G(\theta)^\intercal$, such as $\Sigma(\theta) = H(\theta)$ where $H(\theta)$ is the Hessian matrix Jastrzębski et al. (2017), $\Sigma(\theta) = C$ where $C$ is some constant matrix Mandt et al. (2015), and $\Sigma(\theta) = I$ where $I$ is the identity matrix Chaudhari & Soatto (2018). However, without *any* additional assumptions, the differential symmetries intrinsic to neural network architectures add fundamental constraints on $\Sigma$.

As we showed in section 3, the gradient of the loss, regardless of the batch, is orthogonal to the generator vector field $\partial_\alpha \psi|_{\alpha=I}$ associated with a symmetry. This implies the stochastic noise must also observe the same property, $\langle -\frac{1}{\sqrt{S}} G(\theta)\xi, \partial_\alpha \psi|_{\alpha=I} \rangle = 0$. In order for this relationship to hold for arbitrary noise[6] $\xi$, then $G(\theta)^\intercal \partial_\alpha \psi|_{\alpha=I} = 0$. In other words, the differential symmetry inherent in neural network architectures projects the noise introduced by stochastic gradients onto low rank subspaces.

**Scale Symmetry with Stochasticity.** In the previous section, we performed our analysis using continuous-time model with deterministic gradients to facilitate calculations, and replaced them with stochastic batch gradients upon discretization when we evaluated the results empirically. Instead, we can also directly model the stochastic noise arising from batch gradient with an Itô process to perform analogous analysis in continuous-time as pointed out by Li et al. (2021) in a recent follow-up work.

First, we assume that the time evolution of network parameters $\theta(t)$ during SGD training is described by the Itô process $d\theta = \mu dt + \sqrt{\frac{\eta}{S}} G(\theta) dW_t$, where

$$\mu = -g - \lambda\theta - \frac{\eta}{2}(Hg + \lambda H\theta) + O(\eta^2) + O(\eta\lambda) + O(\lambda^2).$$

Assuming a set of parameters $\mathcal{A}$ respect scale symmetry, then applying Itô's lemma to the function $|\theta_{\mathcal{A}}(t)|^2$ yields the Itô process,

$$d|\theta_{\mathcal{A}}(t)|^2 = \left\{ 2\theta_{\mathcal{A}}^\intercal \mu + \frac{\eta}{S} \text{Tr}[G(\theta_{\mathcal{A}})^T G(\theta_{\mathcal{A}})] \right\} dt + 2\sqrt{\frac{\eta}{S}} \theta^\intercal G(\theta_{\mathcal{A}}) dW_t$$

$$= \left\{ -2\lambda|\theta_{\mathcal{A}}|^2 - \eta|g_{\mathcal{A}}|^2 + \frac{\eta}{S} \text{Tr}[G(\theta_{\mathcal{A}})^T G(\theta_{\mathcal{A}})] \right\} dt.$$

Notice this is a deterministic ODE equivalent to the previously derived ODE with an additional forcing term accounting for the variance of the noise. We can perform an analogous analysis for the case of translation and rescale symmetry.

We can also consider the effect of stochasticity without the complexity of stochastic calculus by considering the dynamics in the discrete setting. As explained in appendix I, this is possible for the case without momentum, but becomes much more complicated once we consider momentum as well.

---

[6]We do not need to assume the noise is Gaussian in order for this property to be true. However, we adopt this commonly accepted assumption to contextualize our work within the literature modeling SGD with an SDE.

## G  DERIVING THE EXACT LEARNING DYNAMICS OF SGD WITH MOMENTUM

We consider a continuous model of SGD with momentum incorporating weight decay (equation 14), momentum (equation 15), stochasticity (equation 31), and modified flow (equation 17). As discussed in appendix C, we can model the effect of momentum by considering the forward Euler discretization $\frac{\theta_{k+1}-\theta_k}{\eta(1-\alpha)}$ and the backward Euler discretization $-\beta\frac{\theta_k-\theta_{k-1}}{\eta(1-\alpha)}$. These terms introduce the numerical artifacts $\frac{\eta(1-\alpha)}{2}\frac{d^2}{dt^2}\theta$ and $\frac{\eta(1-\alpha)}{2}\beta\frac{d^2}{dt^2}\theta$ respectively, as explained by the modified flow analysis in appendix D. Incorporating these elements with weight decay gives the equation of learning,

$$\frac{\eta(1-\alpha)}{2}(1+\beta)\frac{d^2}{dt^2}\theta + (1-\beta)\frac{d}{dt}\theta + \lambda\theta = -g(\theta).$$

Incorporating the effect of stochasticity into this equation of learning gives the Langevin equation,

$$\frac{\eta(1-\alpha)}{2}(1+\beta)dv = -(1-\beta)vdt - \lambda\theta dt - g(\theta)dt + \sqrt{\frac{\eta}{S}}G(\theta)dW_t,$$

$$d\theta = vdt,$$

(32)

where $W_t$ is a standard Weiner process. Compared to the modified loss route (described in the previous section), it is much more natural and simple to account for the effect of stochasticity with modified flow.

**Translation dynamics.** Combining the geometric properties of gradient ($\langle g, \mathbb{1}_\mathcal{A}\rangle = 0$), Hessian ($H\mathbb{1}_\mathcal{A} = 0$), and stochasticity ($G(\theta)^\intercal\mathbb{1}_\mathcal{A} = 0$) introduced by translation symmetry with the Langevin equation of learning (equation 32) gives the differential equation,

$$\left(\frac{\eta(1-\alpha)}{2}(1+\beta)\frac{d^2}{dt^2} + (1-\beta)\frac{d}{dt} + \lambda\right)\langle\theta, \mathbb{1}_\mathcal{A}\rangle = 0.$$

(33)

This is the differential equation for a harmonic oscillator with $\gamma = \frac{1-\beta}{\eta(1-\alpha)(1+\beta)}$ and $\omega = \sqrt{\frac{2\lambda}{\eta(1-\alpha)(1+\beta)}}$. Assuming the initial condition $\frac{d}{dt}\langle\theta(t), \mathbb{1}_\mathcal{A}\rangle\big|_{t=0} = 0$, then the general solution (as derived in appendix H) is

$$\langle\theta(t), \mathbb{1}_\mathcal{A}\rangle = \begin{cases} e^{-\gamma t}\left(\cosh\left(\sqrt{\gamma^2-\omega^2}t\right) + \frac{\gamma}{\sqrt{\gamma^2-\omega^2}}\sinh\left(\sqrt{\gamma^2-\omega^2}t\right)\right)\langle\theta(0), \mathbb{1}_\mathcal{A}\rangle & \gamma > \omega \\ e^{-\gamma t}(1+\gamma t)\langle\theta(0), \mathbb{1}_\mathcal{A}\rangle & \gamma = \omega \\ e^{-\gamma t}\left(\cos\left(\sqrt{\omega^2-\gamma^2}t\right) + \frac{\gamma}{\sqrt{\omega^2-\gamma^2}}\sin\left(\sqrt{\omega^2-\gamma^2}t\right)\right)\langle\theta(0), \mathbb{1}_\mathcal{A}\rangle & \gamma < \omega \end{cases}$$

(34)

**Scale dynamics.** Combining the geometric constraints on the gradient ($\langle g, \theta_\mathcal{A}\rangle = 0$), Hessian ($H\theta_\mathcal{A} = -g_\mathcal{A}$), and stochasticity ($G(\theta_\mathcal{A})^\intercal\theta_\mathcal{A} = 0$) introduced by scale symmetry with the Langevin equation of learning (equation 32) gives the following differential equation[7],

$$\left(\frac{\eta(1-\alpha)(1+\beta)}{4}\frac{d^2}{dt^2} + \frac{(1-\beta)}{2}\frac{d}{dt} + \lambda\right)|\theta_\mathcal{A}|^2 = \frac{\eta(1-\alpha)(1+\beta)}{2}\left|\frac{d\theta_\mathcal{A}}{dt}\right|^2,$$

(35)

This is the differential equation for a driven harmonic oscillator with $\gamma = \frac{1-\beta}{\eta(1-\alpha)(1+\beta)}$, $\omega = \sqrt{\frac{4\lambda}{\eta(1-\alpha)(1+\beta)}}$, and $f(t) = 2\left|\frac{d\theta_\mathcal{A}}{dt}\right|^2$. Assuming the initial condition $\frac{d}{dt}|\theta_\mathcal{A}|^2\big|_{t=0} = 0$, then the general solution (derived in appendix H) is

$$|\theta_\mathcal{A}(t)|^2 = \begin{cases} |\theta_\mathcal{A}(t)|_h^2 + \int_0^t e^{-\gamma(t-\tau)}\left(\frac{\sinh\left(\sqrt{\gamma^2-\omega^2}(t-\tau)\right)}{\sqrt{\gamma^2-\omega^2}}\right)2\left|\frac{d\theta_\mathcal{A}}{dt}(\tau)\right|^2 d\tau & \gamma > \omega \\ |\theta_\mathcal{A}(t)|_h^2 + \int_0^t e^{-\gamma(t-\tau)}(t-\tau)2\left|\frac{d\theta_\mathcal{A}}{dt}(\tau)\right|^2 d\tau & \gamma = \omega \\ |\theta_\mathcal{A}(t)|_h^2 + \int_0^t e^{-\gamma(t-\tau)}\left(\frac{\sin\left(\sqrt{\omega^2-\gamma^2}(t-\tau)\right)}{\sqrt{\omega^2-\gamma^2}}\right)2\left|\frac{d\theta_\mathcal{A}}{dt}(\tau)\right|^2 d\tau & \gamma < \omega \end{cases}$$

(36)

---

[7]The derivation of this ODE uses $\langle\frac{d^2\theta_\mathcal{A}}{dt^2}, \theta_\mathcal{A}\rangle = \frac{1}{2}\frac{d^2}{dt^2}|\theta_\mathcal{A}|^2 - |\frac{d\theta_\mathcal{A}}{dt}|^2$.

where $|\theta_{\mathcal{A}}(t)|_h^2$ is the solution to the homogeneous harmonic oscillator

$$|\theta_{\mathcal{A}}(t)|_h^2 = \begin{cases} e^{-\gamma t}\left(\cosh\left(\sqrt{\gamma^2 - \omega^2}t\right) + \frac{\gamma}{\sqrt{\gamma^2 - \omega^2}}\sinh\left(\sqrt{\gamma^2 - \omega^2}t\right)\right)|\theta_{\mathcal{A}}(0)|^2 & \gamma > \omega \\ e^{-\gamma t}(1 + \gamma t)|\theta_{\mathcal{A}}(0)|^2 & \gamma = \omega \\ e^{-\gamma t}\left(\cos\left(\sqrt{\omega^2 - \gamma^2}t\right) + \frac{\gamma}{\sqrt{\omega^2 - \gamma^2}}\sin\left(\sqrt{\omega^2 - \gamma^2}t\right)\right)|\theta_{\mathcal{A}}(0)|^2 & \gamma < \omega \end{cases}$$

$$(37)$$

**Rescale dynamics.** Combining the geometric properties of gradient $\langle g, \theta_{\mathcal{A}_1} - \theta_{\mathcal{A}_2}\rangle = 0$, Hessian $(H(\theta_{\mathcal{A}_1} - \theta_{\mathcal{A}_2}) + g_{\mathcal{A}_1} - g_{\mathcal{A}_2} = 0)$, and stochasticity $(G(\theta_{\mathcal{A}_1})^\mathsf{T}\theta_{\mathcal{A}_1} - G(\theta_{\mathcal{A}_2})^\mathsf{T}\theta_{\mathcal{A}_2} = 0)$ introduced by rescale symmetry with the Langevin equation of learning (equation 32) gives the differential equation,

$$\left(\frac{\eta(1-\alpha)(1+\beta)}{4}\frac{d^2}{dt^2} + \frac{(1-\beta)}{2}\frac{d}{dt} + \lambda\right)\left(|\theta_{\mathcal{A}_1}|^2 - |\theta_{\mathcal{A}_2}|^2\right) = \frac{\eta(1-\alpha)(1+\beta)}{2}\left(\left|\frac{d\theta_{\mathcal{A}_1}}{dt}\right|^2 - \left|\frac{d\theta_{\mathcal{A}_2}}{dt}\right|^2\right).$$

$$(38)$$

This is the same harmonic oscillator given by equation (35) with the different forcing term $f(t) = 2\left(|\frac{d\theta_{\mathcal{A}_1}}{dt}|^2 - |\frac{d\theta_{\mathcal{A}_2}}{dt}|^2\right)$. The general solution is given by equation (36) and (37) replacing $|\theta|^2$ and $|\frac{d\theta}{dt}|^2$ by $|\theta_{\mathcal{A}_1}|^2 - |\theta_{\mathcal{A}_2}|^2$ and $|\frac{d\theta_{\mathcal{A}_1}}{dt}|^2 - |\frac{d\theta_{\mathcal{A}_2}}{dt}|^2$ respectively.

# H   GENERAL SOLUTIONS FOR ODES

## H.1   EXPONENTIAL GROWTH

Here we will solve for the general solution of the homogenous first-order linear differential equation,

$$\left(\frac{d}{dt} + \lambda\right)x(t) = 0.$$

Assume a solution of the form $x(t) = e^{\alpha t}$. Plugging this in gives the auxiliary equation $\alpha + \lambda = 0$. Thus, the general solution to the differential equation with initial condition $x(0)$ is

$$x(t) = e^{-\lambda t}x(0).$$

Now we will solve the inhomogenous differential equation,

$$\left(\frac{d}{dt} + \lambda\right)x(t) = f(t).$$

Multiply both sides by $e^{\lambda t}$ and factor the left hand side using the product rule such that the differential equation simplies to $\frac{d}{dt}\left(e^{\lambda t}x(t)\right) = e^{\lambda t}f(t)$. Integrate this equation and using the fundamental theorem of calculus rearrange to get the solution

$$x(t) = e^{-\lambda t}x(0) + \int_0^t e^{-\lambda(t-\tau)}f(\tau)d\tau.$$

## H.2   HARMONIC OSCILLATOR

Here we will solve the general solution for a harmonic oscillator,

$$\left(\frac{d^2}{dt^2} + 2\gamma\frac{d}{dt} + \omega^2\right)x(t) = 0.$$

Assume a solution of the form $x(t) = e^{\alpha t}$. Plugging this in gives the auxiliary equation $\alpha^2 + 2\gamma\alpha + \omega^2 = 0$ with solutions $\alpha_\pm = -\gamma \pm \sqrt{\gamma^2 - \omega^2}$. Thus, the general solution to the oscillator equation with initial conditions $x(0)$ and $\frac{dx}{dt}(0) = 0$ is

$$x(t) = e^{-\gamma t}\left(C_1 e^{\sqrt{\gamma^2 - \omega^2}t} + C_2 e^{-\sqrt{\gamma^2 - \omega^2}t}\right)$$

where $C_1, C_2$ are constants

$$C_1 = \frac{\gamma + \sqrt{\gamma^2 - \omega^2}}{2\sqrt{\gamma^2 - \omega^2}}x(0), \qquad C_2 = \frac{-\gamma + \sqrt{\gamma^2 - \omega^2}}{2\sqrt{\gamma^2 - \omega^2}}x(0).$$

Using hyperbolic functions the solution simplifies as

$$x(t) = e^{-\gamma t}\left(\cosh\left(\sqrt{\gamma^2 - \omega^2}t\right) + \frac{\gamma}{\sqrt{\gamma^2 - \omega^2}}\sinh\left(\sqrt{\gamma^2 - \omega^2}t\right)\right)x(0).$$

The form of this general solution implicitly assumes $\gamma > \omega$, the overdamped setting. When $\gamma = \omega$, the critically damped setting, then the solution reduces to

$$x(t) = e^{-\gamma t}(C_1 + C_2 t),$$

where $C_1 = x(0)$ and $C_2 = \gamma x(0)$. When $\gamma < \omega$, the underdamped setting, then the solution reduces to

$$x(t) = e^{-\gamma t}\left(C_1 \cos\left(\sqrt{\omega^2 - \gamma^2}t\right) + C_2 \sin\left(\sqrt{\omega^2 - \gamma^2}t\right)\right),$$

where $C_1 = x(0)$ and $C_2 = \frac{\gamma}{\sqrt{\omega^2 - \gamma^2}}x(0)$.

## H.3 DRIVEN HARMONIC OSCILLATOR

Here we will solve the general solution for a driven harmonic oscillator,

$$\left(\frac{d^2}{dt^2} + 2\gamma\frac{d}{dt} + \omega^2\right)x(t) = f(t).$$

First notice that if $x_h(t)$ is a solution to the homogenous harmonic oscillator and $x_d(t)$ a specific solution to the driven harmonic oscillator, then

$$x(t) = x_h(t) + x_d(t),$$

is the general solution to the driven harmonic oscillator. We will use the Fourier transform to solve for $x_d(t)$.

Let $\hat{x}_d(\tau) = (\nabla\widetilde{\mathcal{L}}x_d)(\tau)$ and $\hat{f}(\tau) = (\nabla\widetilde{\mathcal{L}}f)(\tau)$ be the Fourier transforms of $x_d(t)$ and $f(t)$ respectively. Applying the Fourier transform to the driven harmonic oscillator equation and rearranging gives

$$\hat{x}_d(\tau) = \left(-\tau^2 + 2\gamma i\tau + \omega^2\right)^{-1}\hat{f}(\tau),$$

which implies by the inverse Fourier transform that $x_d(t)$ is the convolution,

$$x_d(t) = (G * f)(t),$$

where $G(t)$ is Green's function (the driven solution $x_d(t)$ for the dirac delta forcing function $\delta_0$),

$$G(t) = \Theta(t)\frac{e^{-\gamma t}}{2\sqrt{\gamma^2 - \omega^2}}\left(e^{\sqrt{\gamma^2 - \omega^2}t} - e^{-\sqrt{\gamma^2 - \omega^2}t}\right),$$

which again using hyperbolic functions simplifies as

$$G(t) = \Theta(t)e^{-\gamma t}\frac{\sinh\left(\sqrt{\gamma^2 - \omega^2}t\right)}{\sqrt{\gamma^2 - \omega^2}}.$$

This form of Green's function is again implicitly assuming $\gamma > \omega$. When $\gamma = \omega$, the function simplifies to

$$G(t) = \Theta(t)e^{-\gamma t}t,$$

and when $\gamma < \omega$, the function simplifies to

$$G(t) = \Theta(t)e^{-\gamma t}\frac{\sin\left(\sqrt{\omega^2 - \gamma^2}t\right)}{\sqrt{\omega^2 - \gamma^2}}.$$

Noticing that both $G$ and $f$ are only supported on $[0, \infty)$, their convolution can be simplified and the general solution for the driven harmonic oscillator is

$$x(t) = x_h(t) + \int_0^t G(t - \tau)f(\tau)d\tau.$$

# I  DERIVING DYNAMICS IN THE DISCRETE SETTING

In section 4 we identified certain parameter combinations associated with network symmetries that are conserved under gradient flow. However, as we explained in section 5, these conservation laws are not observed empirically. To remedy this discrepancy we constructed more realistic continuous models for SGD, incorporating weight decay, momentum, stochasticity, and finite learning rates. In section 6 we derived the exact dynamics for the parameter combinations under this more realistic setting, demonstrating near perfect alignment with the empirical dynamics. *What would happen if we instead derived the dynamics for the parameter combinations directly in the discrete setting of SGD?* Here, we will identify these discrete dynamics and discuss the relationship between the discrete equations and the continuous solutions.

Gradient descent with learning rate $\eta$ and weight decay constant $\lambda$ is given by the update equation $\theta^{(n+1)} = (1 - \eta\lambda)\theta^{(n)} - \eta g(\theta^{(n)})$, and initial condition $\theta^{(0)}$. Using this update equation the sum of the parameters after $n + 1$ steps can be "unrolled" as,

$$\langle \theta^{(n+1)}, \mathbb{1}_{\mathcal{A}} \rangle = (1 - \eta\lambda)^{n+1}\langle \theta^{(0)}, \mathbb{1}_{\mathcal{A}} \rangle + \eta \sum_{i=0}^{n} (1 - \eta\lambda)^{n-i}\langle g(\theta^{(i)}), \mathbb{1}_{\mathcal{A}} \rangle.$$

Similarly, the squared Euclidean norm of the parameters after $n + 1$ steps can be "unrolled" as,

$$|\theta_{\mathcal{A}}^{(n+1)}|^2 = (1-\eta\lambda)^{2(n+1)}|\theta_{\mathcal{A}}^{(0)}|^2 + \eta^2 \sum_{i=0}^{n}(1-\eta\lambda)^{2(n-i)}|g_{\mathcal{A}}(\theta^{(i)})|^2 - 2\eta \sum_{i=0}^{n}(1-\eta\lambda)^{2(n-i)+1}\langle g(\theta^{(i)}), \theta_{\mathcal{A}}^{(i)}\rangle.$$

Combining these unrolled equations, with the gradient properties of symmetry discussed in section 3, gives the discrete dynamics for the parameter combinations,

**Translation:**  $\left\langle \theta^{(n)}, \mathbb{1}_{\mathcal{A}} \right\rangle = (1 - \eta\lambda)^n \left\langle \theta^{(0)}, \mathbb{1}_{\mathcal{A}} \right\rangle$  (39)

**Scale:**  $\left|\theta_{\mathcal{A}}^{(n)}\right|^2 = (1 - \eta\lambda)^{2n}\left|\theta_{\mathcal{A}}^{(0)}\right|^2 + \eta^2 \sum_{i=0}^{n-1}(1 - \eta\lambda)^{2(n-1-i)}\left|g_{\mathcal{A}}^{(i)}\right|^2$  (40)

**Rescale:**  $\left|\theta_{\mathcal{A}_1}^{(n)}\right|^2 - \left|\theta_{\mathcal{A}_2}^{(n)}\right|^2 =$  (41)

$$(1 - \eta\lambda)^{2n}\left(\left|\theta_{\mathcal{A}_1}^{(0)}\right|^2 - \left|\theta_{\mathcal{A}_2}^{(0)}\right|^2\right) + \eta^2 \sum_{i=0}^{n-1}(1 - \eta\lambda)^{2(n-1-i)}\left(\left|g_{\mathcal{A}_1}^{(i)}\right|^2 - \left|g_{\mathcal{A}_2}^{(i)}\right|^2\right)$$

Notice the striking similarity between the continuous equations (18), (19), (20) presented in section 6 and the discrete equations (39), (40), (41). The exponential function with decay rate $\lambda$ from the continuous solutions are replaced by a power of the base $(1 - \eta\lambda)$ in the discrete setting. The integral of exponentially weighted gradient norms from the continuous solutions are replaced by a Riemann sum of power weighted gradient norms in the discrete setting. This is further confirmation that the continuous solutions we derived, and the modified gradient flow equation of learning used, well approximate the actual empirics. While the equations derived in the discrete setting remove any uncertainty about the exactness of the theoretical predictions, they provide limited qualitative understanding for the empirical learning dynamics. This is especially true if we consider the learning dynamics with momentum. In this setting, the process of "unrolling" is much more complicated and the harmonic nature of the empirics, easily derived in the continuous setting, is hidden in the discrete algebra.

## J    EXPERIMENTAL DETAILS

An open source version of our code, used to generate all the figures in this paper, is available at github.com/danielkunin/neural-mechanics.

**Dataset.** While we ran some initial experiments on Cifar-100, the dataset used in all the empirical figures in this documents was Tiny Imagenet. It is used for image categorization an consists of 100,000 training images at a resolution of $64 \times 64$ spanning 200 classes.

**Model.** We use standard VGG-16 models for all out experiments with the following modifications:

- The last three fully connected layers at the end have been adjusted for an input at the Tiny ImageNet resolution ($64 \times 64$) and thus consist of $2048, 2048$, and $200$ layers respectively.

- In addition to the standard arrangement of conv layers for the VGG-16, we consider a variant where we add a batch normalization layer between every convolutional layer and its activation function.

**Training hyperparameters.** Certain hyperparameters were varied during training. Below we outline the combinations explored. All models were initialized using Kaiming Normal, and no learning rate drops or warmup were used.

| Model | Dataset | Epochs | Batch size | Opt. | LR | Mom. | WD | Damp. |
|---|---|---|---|---|---|---|---|---|
| VGG-16 | Tiny ImageNet | 100 | 256 | SGD | $[0.1, 0.01]$ | - | $[0, 0.001, 0.0005, 0.0001]$ | 0 |
| VGG-16 w/BN | Tiny ImageNet | 100 | 256 | SGD | $[0.1, 0.01]$ | - | $[0, 0.001, 0.0005, 0.0001]$ | 0 |
| VGG-16 | Tiny ImageNet | 100 | 128 | SGDM | 0.1 | $[0, 0.9, 0.99]$ | $[0, 0.001, 0.0005, 0.0001]$ | 0 |
| VGG-16 w/BN | Tiny ImageNet | 100 | 128 | SGDM | 0.1 | $[0, 0.9, 0.99]$ | $[0, 0.001, 0.0005, 0.0001]$ | 0 |

Table 4: **Training hyperparameters.**

**Counting number of symmetries.** Here we explain how to count the number of symmetries a VGG-16 model contains.

- Scale symmetries appear once per channel at every layer preceding a batch normalization layer. For our VGG-16 model with batch norm: $2 \cdot 64 + 2 \cdot 128 + 3 \cdot 256 + 3 \cdot 512 + 3 \cdot 512 + 3 = 4,227$.

- Rescale symmetries appear once per channel where there are afine transforms between layers as well as once per input neuron to a fully connected layer. Note that the sizes of the fully connected layers depend on input image size and the number of classes. For our VGG-16 model with and without batchnorm on Tiny ImageNet, this is: $(2 \cdot 64 + 2 \cdot 128 + 3 \cdot 256 + 3 \cdot 512 + 3 \cdot 512 + 3) + (2048 + 1024 + 1024) = 8,323$.

- Translation symmetries appear once per input value to the softmax function, which for the case of classification is always equal to the number of classes, plus the bias term. For our case this is $200 + 1 = 201$.

| | # Params | # Scale | # Rescale | # Translation |
|---|---|---|---|---|
| VGG-16 on Tiny ImageNet | 18,067,464 | 0 | 8,323 | 201 |
| VGG-16 w/BN on Tiny ImageNet | 18,075,912 | 4,227 | 8,323 | 201 |

Table 5: **Counting the number of symmetries.**

**Computing the theoretical predictions.** Some of the expressions shown in equations (18), (19), and (20) in section 6 are not trivial to compute as they involve an integral of an exponentially weighted gradient term. To tackle this problem, we wrote custom optimizers in PyTorch with additional buffers to approximate the integral via a Riemann sum. At every update step, the argument in the integral term was computed from the batch gradients, scaled appropriately, and was accumulated in the buffer. Note that the above sum needs to be scaled by the learning rate, which is the coarseness of the grid of this Riemann sum. Checkpoints of the model and optimizer states were stored at

pre-defined frequencies during training. Our visualizations involve computing the right hand side of equations (18), (19), and (20) from the model states and the left hand side of the same equations from the integral buffers stored in the optimizer states as explained above. These two quantities are referred to as "empirical" and "theoretical" in the figures and are depicted with solid color lines and dotted lines, respectively.

## J.1 ADDITIONAL EMPIRICS

In this work we made exact predictions for the dynamics of combinations of parameters during training with SGD. Importantly, these predictions were made at the neuron level, but could be aggregated for each layer. Here we plot our predictions at both layer and neuron levels for VGG-16 models trained on Tiny ImageNet.

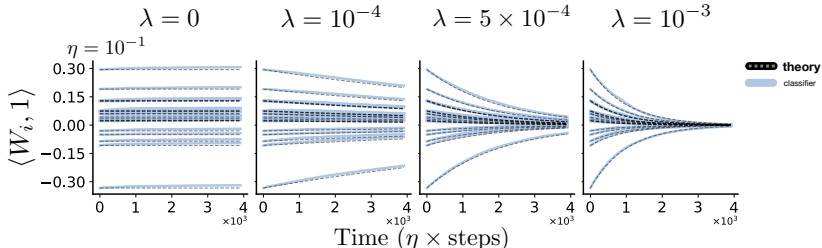

Figure 8: **The planar dynamics of VGG-16 on Tiny ImageNet.** We plot the column sum of the final linear layer of a VGG-16 model (without batch normalization) trained on Tiny ImageNet with SGD with learning rate $\eta = 0.1$, weight decay $\lambda \in \{0, 10^{-4}, 5 \times 10^{-4}, 10^{-3}\}$, and batch size $S = 256$. Colored lines are empirical column sums of the last layer through training and black dashed lines are the theoretical predictions of equation (18).

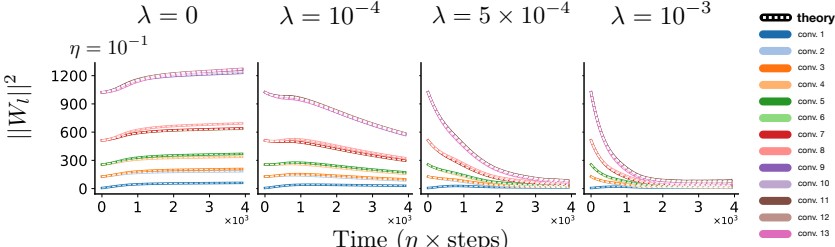

Figure 9: **The spherical dynamics of VGG-16 BN on Tiny ImageNet.** We plot the squared Euclidean norms for convolutional layers of a VGG-16 model (with batch normalization) trained on Tiny ImageNet with SGD with learning rate $\eta = 0.1$, weight decay $\lambda \in \{0, 10^{-4}, 5 \times 10^{-4}, 10^{-3}\}$, and batch size $S = 256$. Colored lines represent empirical layer-wise squared norms through training and the white dashed lines the theoretical predictions given by equation (19).

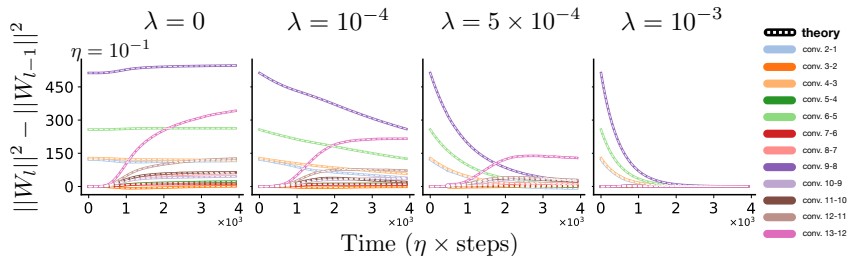

Figure 10: **The hyperbolic dynamics of VGG-16 on Tiny ImageNet.** We plot the difference between the squared Euclidean norms for consecutive convolutional layers of a VGG-16 model (without batch normalization) trained on Tiny ImageNet with SGD with learning rate $\eta = 0.1$, weight decay $\lambda \in \{0, 10^{-4}, 5 \times 10^{-4}, 10^{-3}\}$, and batch size $S = 256$. Colored lines represent empirical differences in consecutive layer-wise squared norms through training and the white dashed lines the theoretical predictions given by equation (20).

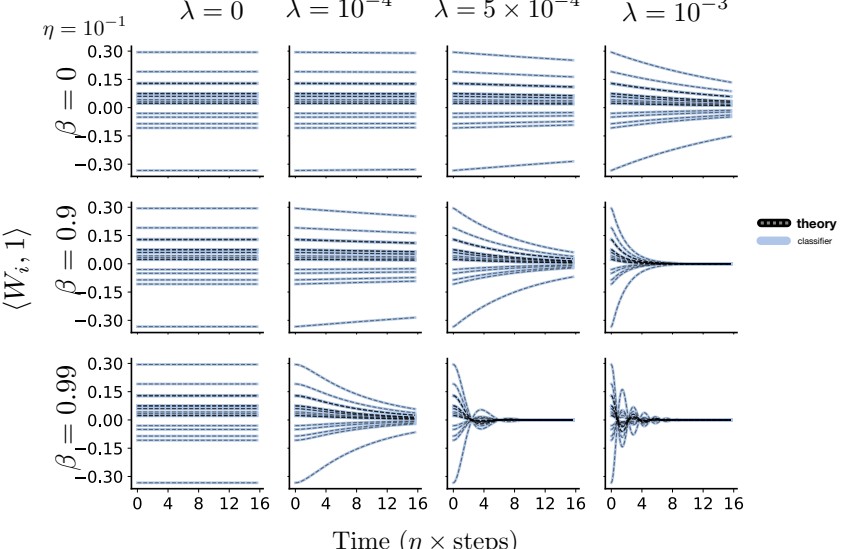

Figure 11: **The planar dynamics of Momentum on VGG-16 on Tiny ImageNet.** We plot the column sum of the final linear layer of a VGG-16 model (without batch normalization) trained on Tiny ImageNet with Momentum with learning rate $\eta = 0.1$, weight decay $\lambda \in \{0, 10^{-4}, 5 \times 10^{-4}, 10^{-3}\}$, momentum coefficient $\beta \in \{0, 0.9, 0.99\}$, and batch size $S = 128$. Colored lines are empirical column sums of the last layer through training and black dashed lines are the theoretical predictions of equation (34).

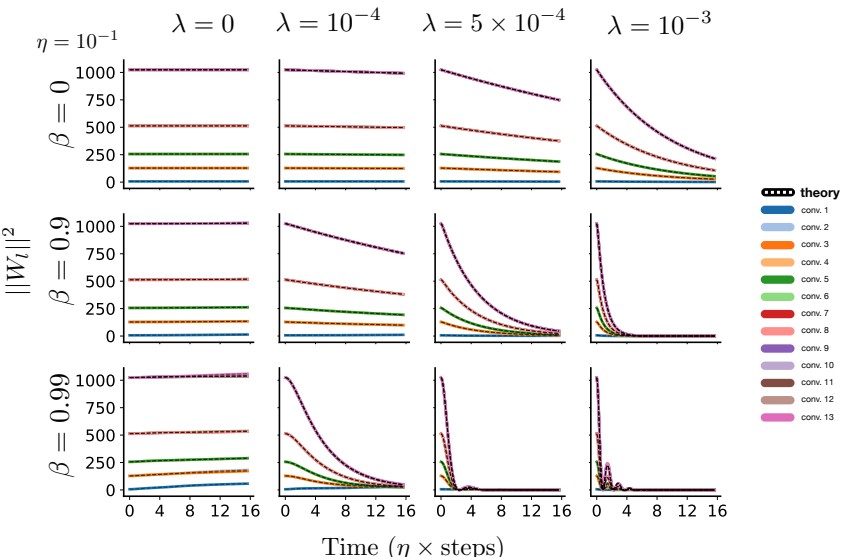

Figure 12: **The spherical dynamics of Momentum on VGG-16 on Tiny ImageNet.** We plot the squared Euclidean norms for convolutional layers of a VGG-16 model (with batch normalization) trained on Tiny ImageNet with Momentum with learning rate $\eta = 0.1$, weight decay $\lambda \in \{0, 10^{-4}, 5 \times 10^{-4}, 10^{-3}\}$, momentum coefficient $\beta \in \{0, 0.9, 0.99\}$, and batch size $S = 128$. Colored lines represent empirical layer-wise squared norms through training and the white dashed lines the theoretical predictions given by equation (36) and (37)

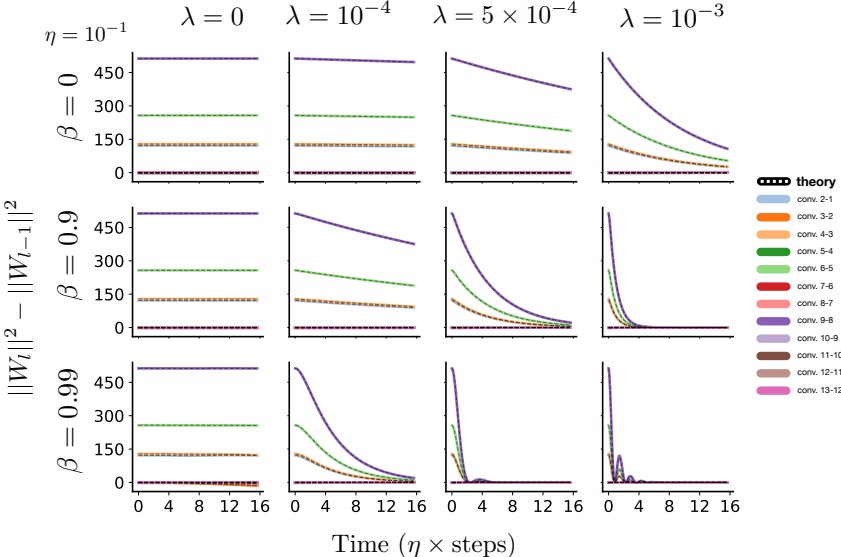

Figure 13: **The hyperbolic dynamics of Momentum on VGG-16 on Tiny ImageNet.** We plot the difference between the squared Euclidean norms for consecutive convolutional layers of a VGG-16 model (without batch normalization) trained on Tiny ImageNet with Momentum with learning rate $\eta = 0.1$, weight decay $\lambda \in \{0, 10^{-4}, 5 \times 10^{-4}, 10^{-3}\}$, momentum coefficient $\beta \in \{0, 0.9, 0.99\}$, and batch size $S = 128$. Colored lines represent empirical differences in consecutive layer-wise squared norms through training and the black dashed lines the theoretical predictions given by equation (36) and (37) replacing $|\theta|^2$ and $|\frac{d\theta}{dt}|^2$ by $|\theta_{\mathcal{A}_1}|^2 - |\theta_{\mathcal{A}_2}|^2$ and $|\frac{d\theta_{\mathcal{A}_1}}{dt}|^2 - |\frac{d\theta_{\mathcal{A}_2}}{dt}|^2$ respectively.

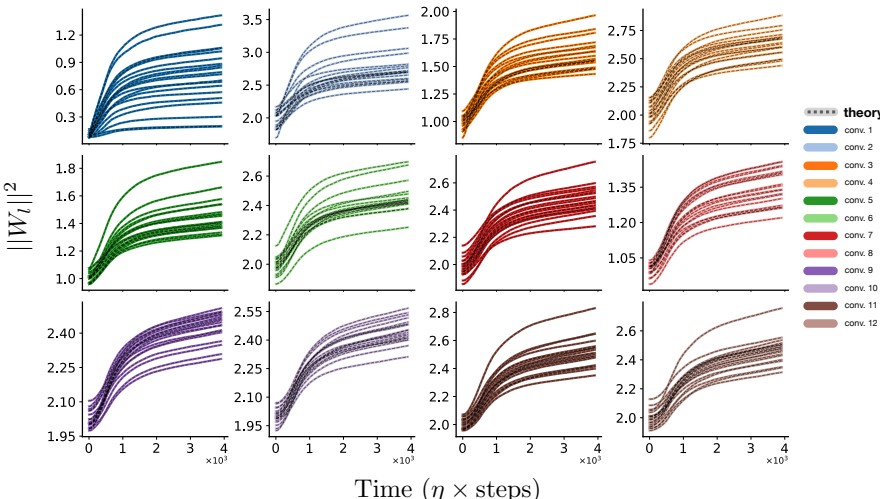

Figure 14: **The per-neuron spherical dynamics of SGD on VGG-16 BN on Tiny ImageNet.** We plot the per-neuron squared Euclidean norms for convolutional layers of a VGG-16 model (with batch normalization) trained on Tiny ImageNet with SGD with learning rate $\eta = 0.1$, weight decay $\lambda = 0$, and batch size $S = 256$. Colored lines represent empirical layer-wise squared norms through training and the black dashed lines the theoretical predictions given by equation (19).

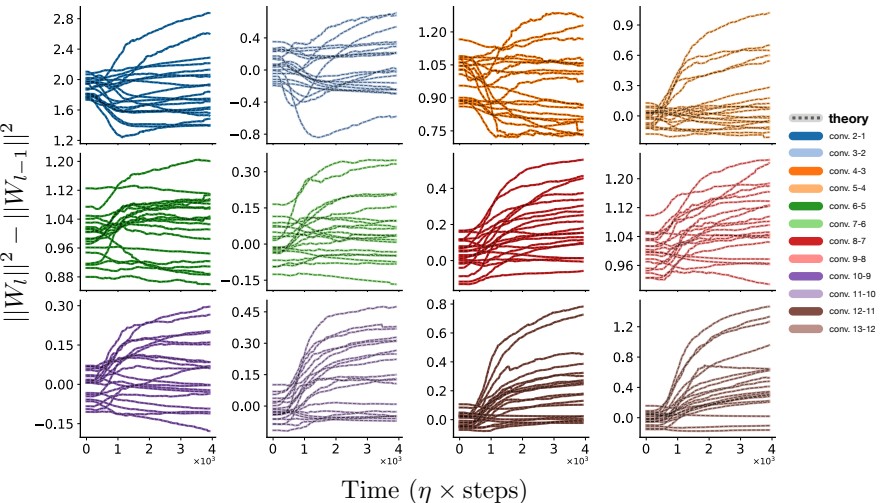

Figure 15: **The per-neuron hyperbolic dynamics of SGD on VGG-16 on Tiny ImageNet.** We plot the per-neuron difference between the squared Euclidean norms for consecutive convolutional layers of a VGG-16 model (without batch normalization) trained on Tiny ImageNet with SGD with learning rate $\eta = 0.1$, weight decay $\lambda = 0$, and batch size $S = 256$. Colored lines represent empirical differences in consecutive layer-wise squared norms through training and the black dashed lines the theoretical predictions given by equation (20).

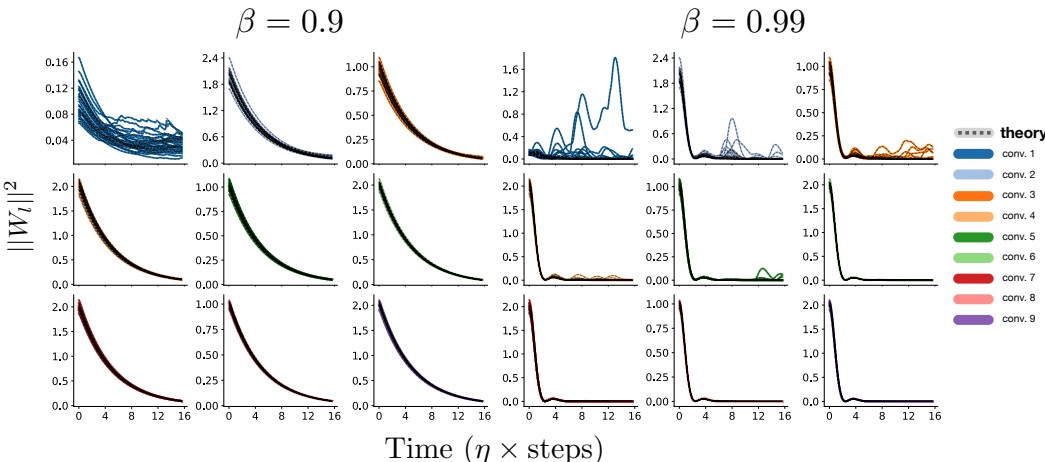

Figure 16: **The per-neuron spherical dynamics of Momentum on VGG-16 on Tiny ImageNet.**
We plot the per-neuron squared Euclidean norms for convolutional layers of a VGG-16 model
(with batch normalization) trained on Tiny ImageNet with Momentum with learning rate $\eta = 0.1$,
weight decay $\lambda = 0$, momentum coefficient $\beta \in \{0.9, 0.99\}$, and batch size $S = 128$. Colored
lines represent empirical layer-wise squared norms through training and the black dashed lines the
theoretical predictions given by equation (36) and (37).

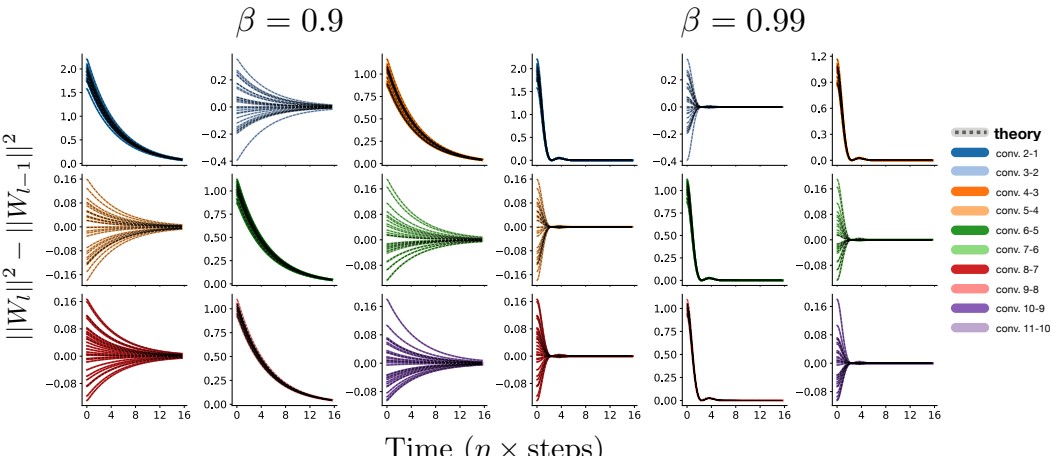

Figure 17: **The per-neuron hyperbolic dynamics of Momentum on VGG-16 on Tiny ImageNet.**
We plot the per-neuron difference between the squared Euclidean norms for consecutive convolutional
layers of a VGG-16 model (without batch normalization) trained on Tiny ImageNet with Momentum
with learning rate $\eta = 0.1$, weight decay $\lambda = 0$, momentum coefficient $\beta \in \{0.9, 0.99\}$, and batch
size $S = 128$. Colored lines represent empirical differences in consecutive layer-wise squared norms
through training and the black dashed lines the theoretical predictions given by equation (36) and
(37) replacing $|\theta|^2$ and $|\frac{d\theta}{dt}|^2$ by $|\theta_{\mathcal{A}_1}|^2 - |\theta_{\mathcal{A}_2}|^2$ and $|\frac{d\theta_{\mathcal{A}_1}}{dt}|^2 - |\frac{d\theta_{\mathcal{A}_2}}{dt}|^2$ respectively.

