# OpenReview forum: "Neural Mechanics: Symmetry and Broken Conservation Laws in Deep Learning Dynamics"
_ICLR.cc/2021/Conference — ICLR 2021 Poster_

### Official Review · AnonReviewer4 · 2020-10-26
**Dynamics of some weight combinations of real world DNN can be predicted**

**Rating:** 7
**Confidence:** 4

**Review:**

The current work studies the implications of continuous symmetries of a DNN on its weight dynamics. Specifically, they consider linearly realized symmetries (such as a shift/translation to the logits), and use the fact that the loss/hessian/mini-batches, are insensitive to such shift to decouple their dynamics. They do so both in the continuous/vanishing-learning-rate case and for small learning rates and manage to provide accurate quantitative predictions for the dynamics of these quantities.

On the positive side, the current work makes accurate predictions on DNNs trained in a real-world setting (real datasets, convolutional layers, finite learning rates, mini-batches etc...) which is a very complicated problem. They do so with a refreshing toolbox, that of symmetries.

On the other hand, what makes their quantities tractable, seems to be the very fact that they have no impact on the DNNs final outputs. For example, this is why they are unaffected by the real-world dataset. While being a clever trick, it can be viewed as an inherent limitation of the approach: one understands quantities that have no bearing on what the DNN learns. I don't see, for instance, how one expects to perform architecture exploration (as mentioned in the discussion), using quantities that have no implications on the DNNs predictions. If the authors can argue that their approach has a broader horizon, it may increase its potential impact.

Two technical comments:

1. I find the use of subset notation and definition of large theta confusing: It seems that small theta is the set of parameters but it is a subgroup of large theta.
2. Inversion symmetry commonly refers to a discrete Z_2 symmetry, whereas the authors take it to be some representation of GL1.

---

> ### Author Response · Authors · 2020-11-18
> **Response to Reviewer 4**
>
> We thank you for clearly recognizing the contribution of our work as “the current work makes accurate predictions on DNNs trained in a real-world setting … which is a very complicated problem”.  To address your technical comments: (1) In section 3 of the updated manuscript, we have introduced a new notation system to discuss subsets of parameters to avoid any confusion. (2) Throughout the updated manuscript, we have replaced all the “inversion symmetry” to “rescale symmetry”. This terminology of “rescale invariance/equivariance” is commonly used in the related work Neyshabur et al. Now we will address your larger remarks:
>
> **The parameter combinations do depend on data.** We apologize for any confusion that we may have created, which lead the reviewer to conclude that parameter combinations that we predict “are unaffected by the real-world datasets.”  While the parameter combinations are conserved under gradient flow, they are *not* conserved at finite learning rate, and the gradient norms can drive changes in these quantities (see equations 19, 20); therefore their learning dynamics are indeed directly driven by the data (except for translation symmetry).  We have significantly revised our discussion in section 6 to highlight this data dependence.
>
> **Symmetry transformations do impact the learning dynamics as the gradient and regularizer need not to respect symmetry.**  You are correct that, by its very definition, the symmetry transformations on parameters have no effect on the outputs and loss of the network. However, these transformations have a crucial impact on the learning dynamics. This is because, even if the network outputs are unaffected under some transformation of the parameters, the gradient and regularization for these parameters can change.  For example, in the case of scale symmetry, if we increase the norm for parameters preceding a batch normalization layer, then the output of the network doesn’t change, but the gradient and L2 regularization for these parameters decrease and increase, respectively.  This property has been used to explain how batch normalization directly controls the effective learning rate for its parameters, impacting their learning dynamics. Therefore, understanding the dynamics of these norms is of crucial importance to understand how the effective learning rate evolves throughout training and thus the network as a whole. Remarkably, our work provides the exact solution for this quantity. Overall, our work provides a theoretically tractable direction for exploring how the training hyperparameters (weight decay, momentum, batch size, learning rate) affect the learning dynamics.
>
> **Principles of symmetry have been and will be crucial in designing new network architectures and optimizers.** Understanding how symmetries in the loss affects learning dynamics through gradient and Hessian geometries has been crucial in designing new network architectures and optimizers. For example, one of the important motivations behind the invention of Batch Normalization by S. Ioffe and C. Szegedy was the realization that the scale invariance of Batch Normalization can stabilize gradient signal propagation while leaving networks’ outputs intact. Similarly, the motivation for the invention of the Path-SGD optimizer by B. Neyshabur et al., was the fact that gradient descent does not respect rescale equivariance even when the network outputs respect rescale invariance. In this work, we have unified and generalized the geometric properties of the gradient and Hessian induced by symmetry (see a new section A in the Supplementary Materials), thus providing a theoretical foundation for principled design of architectures and optimizers to achieve certain geometric goals.
>
> Thank you again for your feedback. Your suggestion to focus on the broader horizon has significantly helped us refine our paper from motivation to interpretation of our theoretical and empirical results. We hope these updates have clarified the potential future impact of our work.

---

> > ### Comment · AnonReviewer4 · 2020-11-22
> > **2nd Response**
> >
> > I thank the referee for their clarifications.
> >
> > The authors have convinced me that the differential equations they derive do depend, in several important cases, on the data via the gradients of the DNN. Unfortunately, at least as far as I can see, in those more advanced cases, these gradients must be obtained empirically. Once the latter are known, the equations given in this work can be used the predict the specific weight combinations ($\Theta$'s). Hence to the extent to which the manuscript allows strictly-analytical predictions on real-world DNN trained on real-world datasets--- the theory/PDE is data agnostic.
> >
> > Please let me know if I missing something here.

---

> > > ### Author Response · Authors · 2020-11-23
> > > **2nd Response to Reviewer 4**
> > >
> > > You are correct, the differential equations and their exact solutions for scale and rescale symmetry (eq 19, 20), under the realistic settings of finite batch size and finite learning rates, depend on the data via the norms of the gradients, which must be obtained empirically (see Supplementary Material G for our implementation details). However, we still believe our analytic understanding of the dynamics of parameter combinations in realistic settings is of value, as we explain below.
> > >
> > > **Defining the relationship between gradient norms, optimization hyperparameters, and learning dynamics.** As discussed in section 4, predicting weight combinations based upon gradient flow leads to data agnostic conservation laws. However, if you are a practitioner who cares about learning dynamics with finite step size, with or without momentum, and with finite batch size, you would soon find that these quantities are *not conserved* and instead present highly complex dynamics (Fig. 1 and Fig. 5).  Prior to our theory, this complex dynamics may seem quite disconcerting to a practitioner. For example, in Fig. 5 the overall scale of weights can go up, go down, or even exhibit nonmonotonic dependence (middle row).  Also in Fig. 5 the difference in weight norms gives complex dynamics (bottom row).
> > >
> > > A practitioner seeing all this complexity might be startled and might wonder: (1) why are these quantities even changing when the gradient flow theory claims they should be conserved? (2) does there exist any simple analytic dynamical law that governs their motion, and if so, what is that law and how do the myriad possible quantities (general data structure, momentum terms, weight decays, batch size, learning rate, etc..) all interact to generate this law; and (3) can this law predict the motion quantitatively?
> > >
> > > Sections 5 and 6 answer these questions definitively, and equation 19 and 20 are the answers.
> > >
> > > **Intriguingly, we show there exist very simple analytic laws that govern the complex dynamics in Fig. 1 and 5.** Moreover, we show that these dynamics, which could have depended on the data in a priori arbitrarily complex ways, *instead depend on the data **only** through the norms of the gradient*.  Thus the situation is simpler than one might have had any reason to initially believe, given the complexity of the dynamics seen in Fig. 1 and Fig. 5.  Of course, to solve these analytic laws of motion, one must integrate these equations with the observed gradient norms.  A nice analogy comes from other spheres of mechanics. For example, the motions of planets are governed by analytic laws, and their discovery was fundamental. But to predict the motion of more than two planets, these analytic laws must still be numerically integrated.  In our setting, we feel these analytic laws of motion are a very useful contribution, as they describe the complex dynamics of thousands of weight combinations in real world networks and datasets, and reduce the dependence on the data only to gradient norms that act through integral terms.  Indeed, these analytic laws suggest the presence of strong oscillations with momentum, **a prediction which we then tested empirically and confirmed**  (Fig. 6 right panel), demonstrating the power of our theory to predict interesting phenomena and the quantitative hyperparameter choices that generate them.
> > >
> > > Most importantly, these analytical laws match empirical results on real-world DNN trained on real-world datasets exactly.  **We feel this is not a trivial accomplishment.**  Understanding the exact functional relationship between the data dependence (and more precisely isolating this data-dependence specifically to  gradient norms), the optimization hyperparameters of batch size, learning rates, and momentum terms,  and the learning dynamics of certain weight combinations is the main contribution of this work.
> > >
> > > Future work could consider how to use this precise understanding to obtain analytic control over the weight combinations prior to even training.  For example, one route might be assuming certain properties of the gradient norm such as it is constant through training or proportional to the weight combination itself.  Another route would be considering how normalizing the gradients or adapting the learning rate might simplify the integral term.  It's worth noting that we empirically compute this integral term by slightly modifying the PyTorch SGD optimizer with an additional buffer keeping an exponentially moving average of gradient norms during training (see Supplementary Material G), which is very similar to the buffers introduced by adaptive learning algorithms such as Adam, Adagrad, and RMSprop.  Understanding the relationship between these analytic laws and adaptive optimization is a promising direction for future work.

---

### Official Review · AnonReviewer1 · 2020-10-26
**Interesting theoretical perspective on dynamics of DNNs but with some confusions**

**Rating:** 6
**Confidence:** 3

**Review:**

This paper analyzes the learning dynamics of DNNs from the perspective of symmetry of some parameters. It is interesting to borrow ideas from physics, which I believe is a right way to go.

Specifically, the paper derives analytical form of parameters under the cases of translation, scale and inversion invariances, and also modified the underlying gradient flows to accommodate stochastic gradients. The results are very interesting, which I like a lot. However, I have some confusions about some results, which make me not able to give the paper a pass at this time. I will consider revising the score if the authors can clear me in the rebuttal.

1. The papers talks about three invariances, and also given some examples of DNNs that satisfies these invariances. From my understanding, these invariances only apply to a very limited cases of DNN structures, e.g., the softmax layer. And in Figure 3-5, the authors verify the convolutional layers for these invariances. It is not clear to me why convolutional layers satisfy these invariances:  translation, scale and inversion. Am I missing something?

2. In 6.1, a new solution for the translation invariance case is derived. However, it seems that the parameters following the solution still converge to zero? This is obviously not the case in practice.

3. Third line below eq.16, it says "there is a competition between the centripetal effect of weight decay and the centrifugal effect of discretization." I don;'t understand why the descretization has the centrifugal effect.


==========
After rebuttal: The rebuttal resolves most of my concerns. It is more clear to me that this is an interesting paper on describing the dynamics of DNN parameters in the training, which seems novel to me. I also realize that the closed form dynamics are not for individual parameters, but in terms of some statistics of the parameters, e.g., the sum of the parameters. This makes the results not as existing as what I thought. That is why I decide to raise my score to 6.

---

> ### Author Response · Authors · 2020-11-18
> **Response to Reviewer 1**
>
> Thank you for your constructive review. We are glad that you like our approach and results, finding them very interesting. We will now respond to your specific questions, hopefully clarifying any confusion we have generated:
>
> 1. **Nearly every parameter in modern neural network architectures is involved in one of the three invariances discussed in our paper.** As we discussed in section 3, at any hidden unit with batch normalization, the scale symmetry applies to the parameters into this neuron. At any hidden unit with a homogeneous activation function ($f(x)$ obeying $f(\alpha x) = \alpha^n f(x)$, for example ReLU obeys when $n=1$) the rescale symmetry applies to the parameters in and out of this neuron. In the case of consecutive convolutional layers, this means the filters into a channel and the filters out of that channel. And as you mentioned, the parameters immediately preceding a softmax function observe translation symmetry. Thus, most parameters in modern deep neural networks are involved in at least one symmetry.  In the case of VGG-16 with batch normalization trained on Tiny-ImageNet (one of the model/dataset combinations we considered in section 6) there are 12,751 distinct symmetries and every single parameter is involved in at least one of these symmetries.  See table 5 in Supplementary Material G for a breakdown of the number/type of symmetries for the models we used.
> 2. **Parameters that respect translational symmetry do not converge to zero, their sum does.** As you can see in equation 18 (section 6), the projection of parameters observing translation symmetry $\theta$ onto the all-ones vectors $\langle \theta, \mathbb{1}_{\mathcal{A}} \rangle$ exponentially converges to zero. Please note that the parameters $\theta$ are not going to zero, their sum is. There are many ways for this projection to converge to zero without the parameters themselves converging to zero, unlike in the case of scale symmetry.
> 3. **Discretization of circular motion leads to a growing norm due to the curvature of the trajectory (see section 6 and Supplementary Material C).** Motivated by your question, we have clarified our explanation in the main text (section 6) and added further intuition in the Supplementary Materials, which we will go over now (see Figure 7 in Supplementary Material C.) Consider a particle moving in circular motion as discussed/depicted in Figure 7. For an infinitesimal step size, then our particle will stay on the circle defined by its initialization.  However, for a finite step size then with each discrete update we “fall off” the circle that we were just on to a circle with a larger radius.  The larger the step size, the greater the change in radius.  Intuitively, we can understand discretization to be leading to a centrifugal effect moving our particle away from the origin. We can formulate these intuitions through modified equation analysis, where we can derive that discretization of these first order dynamics leads to a negative acceleration term, countering the acceleration into the origin needed for circular motion. We have further expanded on the modified equation analysis in the Supplementary Material in the updated manuscript.
>
> We sincerely hope that our reply has clarified your confusions, and you could now revise the score in light of the updated manuscript and the detailed discussion above.  However, please let us know if you have any further questions.

---

> > ### Comment · AnonReviewer1 · 2020-11-23
> > **about symmetry**
> >
> > Thanks for the response. Can you confirm that if there is no batch norm layer, what types of symmetry does the convolutional operator follow?

---

> > > ### Author Response · Authors · 2020-11-23
> > > **Follow Up to Reviewer 1**
> > >
> > > Without batch normalization then sequential convolutional operators will respect **rescale symmetry** (assuming the activation function is homogeneous).  To understand this imagine  scaling the entire filter associated with a channel by alpha and then at the next convolutional layer scaling the components associated with that channel in all filters by 1/alpha.  If the activation function “in between” the convolutional  operators is homogeneous, then the output of the function will be unmodified as these scales will cancel.  This is the same argument for fully-connected networks where we consider rows and columns of sequential linear operators.  There will be as many rescale symmetries as shared channels between the two operators.

---

> > > > ### Comment · AnonReviewer1 · 2020-11-25
> > > > **thanks for the clarification**
> > > >
> > > > I have raised my score to 6. The reason is partly because that I realize that the closed form dynamics are not for individual parameters, but in terms of some statistics of the parameters, e.g., the sum of the parameters. This makes the results not as existing as what I thought. Overall, this is a good paper.

---

### Official Review · AnonReviewer3 · 2020-10-26
**Clear paper but no significant contribution**

**Rating:** 5
**Confidence:** 3

**Review:**

This paper studies the dynamics of the parameters while training a neural network via SGD. SGD is not studied directly but three continuous time approximations of SGD are considered: the classical gradient flow, a stochastic differential equation of Langevin type, and a "modified" gradient flow whose derivation has roots in the literature. For each dynamics, the authors show that some invariant properties of the loss function (which are often satisfied in practice) imply some invariant quantities for the dynamics.

Overall, the paper is rather clear. It tries to provide a physical meaning behind the dynamics of learning, which is an interesting question.

However, I do not see any significant theoretical contribution. For instance, all derivations are simple differential calculus applications. Moreover, several models for SGD are used, each of them have their own invariant properties (which look alike) and then what? The technical contribution is not clear to me. Finally, the experimental contribution is rather mild because the numerical experiments are not discussed in the main text (except marginally in the conclusion). There is a lot of room for improvement (see below) and I don't recommend the paper for publication in this form.


Specific remarks:
- The first paragraph of Section 3 is not necessary in my opinion (already explained in the intro). Moreover, the notation for the group action is a bit misleading, e.g. there is confusion between \psi, \psi(\theta) and \psi(\theta,\alpha)
- Eq 2: the translation invariant suddenly applies only to a subset of the parameters.
- Figure 1, 2 are not commented in the main text. Same for 3,4,5 (except in the conclusion)
- The authors could recall Noether's theorem for comparison
- The models used for SGD are not theoretically justified, except for the classical gradient flow, for which it is standard (Kushner & Yin 2003). For Eq (11) it starts to be sloppy (CLT + Forward Euler). For Eq (13), only intuition is provided.
- Last paragraph of Page 5. It seems that the discussion applies for any \xi (not necessarily Gaussian). Does it help to remove the Gaussian noise assumption?





To improve the paper, I suggest the authors to
- clearly locate their work within the existing literature. This would help to understand why the questions answered in this paper are important, and to highlight their contribution.
- Turn their paper into an experimental one. To this end, dedicate a whole section to **commented** numerical experiments. This does not mean adding more simulations, just explain them in the main text and how they contribute to the main message of the paper.


Minor:

Page 2: "nornalization"
Between Eq 3 and 4: Notation not defined
Page 4: ReLU not defined (give the formula)
Section 5.1, 2nd line. \nabla is missing
Page 6: graident
Page 7 (two times): previosly

---

> ### Author Response · Authors · 2020-11-18
> **Response to Reviewer 3 (1/2)**
>
> Thank you for your detailed reviews and constructive suggestions which have reshaped our manuscript significantly. We are glad that you find our question interesting and think our presentation is clear, but with “room for improvement”. We have incorporated all of your specific remarks and minor comments, which you can find in the updated manuscript.  We will now  explain how we have incorporated your two major suggestions about (i) locating our work within broader context and (ii) creating a new section in the main text focusing on the empirics.
>
> ### Locating our work in the existing literature
> To better locate our work within the existing literature, we have (i) intensively edited the introduction, (ii) added a whole new discussion reviewing existing literature of learning dynamics, (iii) created a new table in section A in the Supplementary Materials presenting how we unify and generalize the existing literature on the geometry of loss landscapes through the lens of symmetry.
>
> **Our work opens up a new direction for understanding training dynamics, by not making simplifying assumptions, but rather restricting our predictions to meaningful parameter combinations (see updated section 1 & 2).**  A foundational question in deep learning theory is: what, if anything, can we quantitatively predict about the training dynamics of large-scale, non-linear neural network models driven by real-world datasets and optimized via stochastic gradient descent with a finite batch size, learning rate, and with or without momentum? In order to make headway on this extremely difficult question, existing works have made major simplifying assumptions on the network, such as restricting to identity activation functions Saxe et al., infinite width layers Jacot et al., or single hidden layers Saad & Solla. Many of these works have also ignored the complexity introduced by stochasticity and discretization by only focusing on the learning dynamics under gradient flow. In the present work, we make the first step in an orthogonal direction. Rather than introducing unrealistic assumptions on the model or learning dynamics, we uncover restricted, but meaningful combinations of parameters with simplified dynamics that can be solved exactly without introducing a single assumption (see the new Fig. 1 in our paper). We make this fundamental contribution by using the tools of symmetry and modified equation analysis, which have been underutilized in deep learning. Our work provides a new arsenal of tools to analyze learning dynamics and evidence that exact predictions are possible even for non-linear architectures, real-world datasets, and learning rules at both finite step sizes, batch sizes, and with or without momentum. For example, in the case of VGG-16 with batch normalization trained on Tiny-ImageNet (one of the model/dataset combinations we considered in section 6) there are 12,751 distinct symmetries which means that we can analytically describe the learning dynamics of 12,751 parameter combinations, again at finite step sizes, batch sizes, and with or without momentum, and we confirm our analytic predictions against numerical simulations. To our knowledge, this has never been done before in any real world setting.
>
> **We have added a whole new section describing how our symmetry-based proof on the geometry of the loss landscapes unifies, simplifies and generalizes existing literature (see Supplementary Material A).** Inspired by your suggestion, we have added a whole new section A in the Supplementary Materials formalizing our derivation of symmetry induced geometric properties and created a new subsection detailing how they relate to the existing literature. First, you are correct that “all derivations are simple differential calculus”. However, we believe that the mathematical simplicity of our proof strategy via symmetry is a strength, not a weakness, just as in Noether’s theorem. Notably, our symmetry-based proofs automatically, in a unified manner, yield as many as 15 formulas for the geometric properties of loss landscapes, in the form of constraints on gradients and Hessians.  Remarkably, to the best of our knowledge, some of the 15 formulas that we derive are new (see table 1 in the Supplementary Materials). Others have been derived in previous literature, but many of their proofs are tedious, algebraic, or sometimes make unnecessary assumptions such as biases can’t be included or the properties only hold per layer. Conversely, our strategy is as simple as, if you can identify a differentiable invariance for a set of parameters, then the geometric properties associated with that invariance hold for those parameters. In the future, we expect that our proof strategy of harnessing symmetry will become the standard when investigating the geometry of the loss landscape. So we believe our general, unifying proof strategy itself is a significant contribution.

---

> ### Author Response · Authors · 2020-11-18
> **Response to Reviewer 3 (2/2)**
>
> ### Strengthening our empirical contributions
>
> **New theoretical and empirical study of stochastic gradient descent “with momentum” (see section 5, 6, and Supplementary Material B, E, H).** Motivated by your constructive feedback, we have generalized our theoretical and empirical results to include stochastic gradient descent with momentum. Momentum is a commonly used hyperparameter when training deep neural networks and is thus a necessary consideration to obtain theory that can empirically match up with large scale real-world deep networks. To expand our analysis we have taken the following approach:  (1) We considered how momentum alone affects the gradient flow ODE to derive a continuous-time limit for momentum.  (2) We demonstrate how modified equation analysis interacts with this limiting differential equation, resulting in a second-order ODE for the momentum dynamics.  (3) We apply this equation of motion with our symmetries to derive exact solutions for the dynamics of meaningful parameter combinations.  Surprisingly, the solutions we obtain take the form of driven harmonic oscillators where the driving force is given by the gradient norms, the friction is defined by the momentum constant, the spring coefficient is defined by the regularization rate, and the mass is defined by the the learning rate and momentum constant. Amazingly, we again find a theoretical parallel between physics and deep learning. For most standard hyperparameter choices, our solutions are in the overdamped setting and align well with our previous first-order solutions up to a time rescaling.  However, for large values of the momentum constant, we can push the solution into the underdamped regime where we would expect harmonic oscillation and indeed, we empirically verify our nontrivial predictions even at scale for VGG-16 trained on Tiny ImageNet.  Our success expanding our original analysis to the setting of momentum also provides a rubric for how future work might consider the impact of adaptive optimizers, such as Adam, Adagrad, RMSprop, or the more recent LARS/LAMB, on the training dynamics of neural networks.
>
> **An updated section dedicated to commented empirics (see section 6, and Supplementary Material H).** We really appreciate your suggestion of how to focus more directly on the empirical results we already have and their qualitative conclusions. We agree that the original form of our numerical experiments was verbose and diluted the main findings and conclusions of our experiments. We have reworked this section, by in part moving most of the derivation of the solution to the appendix and rather focusing on the commented experiments in section 6 in the main text. We invite you to re-read this section and hope that this addresses your constructive feedback. While space constrained us from further discussing our empirics in the main paper, nevertheless, we added a new section H in the Supplementary Materials presenting all of our experimental findings per layer (Fig. 8,9,10,11,12,13) and per neuron (Fig. 14,15,16,17).
>
>
> We would like to thank you again for your very helpful feedback that has significantly improved our paper. We sincerely hope that these major revisions have now addressed all of your concerns.

---

> > ### Comment · AnonReviewer3 · 2020-11-23
> > **Update**
> >
> > Thank you for updating the paper and your detailed answers. I am quite satisfied with most of them but in my opinion, the paper still lacks a theoretical justification for the continuous time dynamics.
> >
> > I will upper my score.

---

> > > ### Author Response · Authors · 2020-11-23
> > > **2nd Response to Reviewer 3**
> > >
> > > We are very glad that you are quite satisfied with most of our paper updates and have raised your score. We thank you very much for your open-mindedness and flexibility.  However, we are surprised that you are still not voting for acceptance based on a single remaining point that "the paper still lacks a theoretical justification for the continuous time dynamics". Indeed, in the updated manuscript, we have dedicated the entire section 5 as well as the sections B and C of Supplementary Material to build up the continuous-time dynamics step by step. If you had not seen the updated versions of Sec. 5 and Sec. B and C, which we intensively wrote to address your concerns, we would greatly appreciate it if you could take a look, and revise your score further upwards if these sections do indeed address your last remaining concern about a theoretical justification for the equations.  If they do not, then if you could be more specific about what your concerns are (i.e. which exact steps, etc…), then we would be more than happy to address your concerns even more effectively.
> > >
> > > Also please note: existing literature theoretically justifies the continuous time dynamics and we only strengthen it through extensive confirmation against numerical experiments. Indeed, as we discussed in section 5, solid mathematical justifications of the modified equation analysis are provided in the existing literature that we properly cite. For example, existing literature that we cite in section 5, “ Li et al. (2017); Feng et al. (2019) and most recently Barrett & Dherin (2020)” for SGD and “Kovachki & Stuart (2019)” for momentum provide a solid theoretical justification for the continuous time dynamics. Additionally, modeling the stochasticity of SGD with a stochastic differential equation as done in section 5 is standard practice in existing literature (see Mandt et al. (2015)). If anything, our paper provides by far the strongest empirical tests of the continuous time dynamics, by considering how they interact with symmetry, which only strengthens the "justification for the continuous time dynamics” on top of the existing literature. Based on our extensive  tests of our equations against numerical simulations in Fig. 5 and 6,  we hope there can be no doubt as to the correctness of the equations.
> > >
> > > We again thank you for your great care in reviewing our manuscript. It has become better and more thorough because of your input.  We hope you will like the new sections we wrote with your concerns in mind.

---

> > > > ### Comment · AnonReviewer3 · 2020-11-24
> > > > **Sections 5, B and C**
> > > >
> > > > I did appreciate the updates on your paper, and that's one of the reasons why I raised my score from 3 to 5.
> > > >
> > > > I would vote for acceptance if symmetries were theoretically studied for SGD directly rather than approximating continuous time dynamics. I understand that the dynamics considered in the paper approximate well the discrete time SGD. But in my experience, dealing with SGD itself is more difficult and can lead to surprises.
> > > >
> > > > In particular,  this modified flow does approximate better SGD in terms of the proximity of the trajectories. Now, the paper doesn't consider how the symmetries will change by considering the discrete time SGD, except in the numerical exp.

---

> > > > > ### Author Response · Authors · 2020-11-25
> > > > > **3rd Response to Reviewer 3**
> > > > >
> > > > > We are glad that you “appreciate the updates” to our paper and that we have fully addressed your original suggestions for improvement. Yet, we are quite surprised that you are still inclined not to accept our paper because of the single fact that we use continuous-time dynamics and that you are now suggesting we rewrite our paper using discrete time dynamics, a comment you did not originally make. This is especially surprising, given the fact that **the existing literature has made significant theoretical progress by using continuous-time dynamics** (e.g., Saxe et al., S.S. Du et al., Jacot et al., Mandt et al.).
> > > > >
> > > > > We understand that predictions from simple continuous time dynamics, such as gradient flow, can deviate largely from the real-world dynamics of SGD. Indeed, this very fact was our motivation to construct a better continuous time dynamics model as thoroughly explained in section 5, Supplementary Material B, and C.  Indeed the dynamics using our realistic continuous-time models, mathematically justified in the existing literature (Li et al. (2017); Feng et al. (2019); Kovachki & Stuart (2019)), lead to “more difficult”, complex dynamics, confirming your “experience, dealing with SGD”, and match thoroughly with empirics from of discrete time finite rate SGD learning dynamics of  state-of-the-art models with millions of parameters trained on large-scale datasets (figures 5, 6, 8, 9, 10, 11, 12, 13, 14, 15, 16, 17). Thus, given our solid theoretical and empirical groundings we would appreciate if you could be clearer where you think “surprises” we overlooked could come from. Mathematically, “surprises” can only arise from incorrect approximations. If you think there might be a “surprise”, point to us which exact approximation we made in the construction of our realistic model of SGD that warns you the most.
> > > > >
> > > > > If by “surprise” you mean the dynamics of SGD are difficult to understand, then we agree.  While it might be “surprising” that the norms of the weight matrices before batch normalization are monotonically increasing through training, it is not surprising once you understand that the dynamics for the norm are driven by an exponentially moving sum of gradient norms. While it might be “surprising” that the column sums of the final layer of a VGG-16 model oscillates between positive and negative values, it is not surprising once you understand that their dynamics are driven by a second-order ODE describing a harmonic oscillator.  It might be equally “surprising” why this behavior arises for certain hyperparameter choices but not others, yet again, this is not surprising when you understand how the hyperparameters of optimization impact the coefficients of the ODE.  **Our work has already made significant progress towards demystifying the “surprising” dynamics of SGD**.
> > > > >
> > > > > You are incorrect when you state that “the paper doesn't consider how the symmetries will change by considering the discrete time SGD”.  As we clearly state in section 3, these symmetries hold “at all points in training, no matter the loss or dataset”.  **Symmetries are inherent properties of a network’s architecture and are completely independent of the optimization process**. If your comment was actually about how the “conservation laws” will change by considering the discrete time SGD, then that’s what the rest of the paper is about,  as you confirmed, “I understand that the dynamics considered in the paper approximate well the discrete time SGD.”

---

### Official Review · AnonReviewer2 · 2020-10-28
**Very nice analysis and accurate predictions using symmetry**

**Rating:** 8
**Confidence:** 3

**Review:**

Pros:
- This paper is very well-written and motivated.
- The train of thoughts is explained very clearly, such that I (admittedly not being  an expert in this field) was able to follow.
- The idea to unify invariances of the loss function by using symmetries and derive corresponding conservation laws (for $\lambda =0$) in the gradient flow is very elegant.
- By adapting a modification of the gradient flow from previous works that accounts for the discrete approximation of SGD, the derived theory was able to predict the behavior of the relevant quantities during training to a remarkable accuracy.

Cons: I did not find any major drawbacks of this work. Just two small questions:
- Using $\ell^2$ regularization on a problem with scale symmetry does not seem to make sense, because the cost function
$$ \mathcal{L}(\theta) + \lambda \|\theta\|^2 $$
will likely not have a minimizer as soon as $\lambda >0$. Reducing the magnitude of any $\theta$ that is optimal for $\mathcal{L}$ reduces the regularization, but in the limit of $\theta=0$ the loss might jump up. Thus, the costs are not lower semi-continuous.
- Additionally, the scale symmetry seems to naturally lead to a discontinuous loss function $\mathcal{L}$. Is there no problem in even defining the gradient flow for such a function? Which properties of $\mathcal{L}$ do you need to derive the continuous gradient flow equations?


Overall, I really enjoyed reading this paper. Since I am not an expert in the field, I cannot really judge the novelty/contribution, but aside from this aspect, I clearly recommend the acceptance of this work.

-----------
- There is a typo in Section 6.1 "graident"
- I stumbled upon the NeurIPS 2020 paper "Reconciling Modern Deep Learning with Traditional Optimization Analyses: The Intrinsic Learning Rate" by Li, Lyu, and Arora. Based on the abstract, this seems to be a relevant related work.


----------------
After the rebuttal: I'd like to thank the authors as well as my fellow reviewers for the interesting discussions and corresponding clarifications. Summarizing my impressions from the discussion, the two main points of criticism are that the proposed analysis is not fully predictive (depends on the norm of the gradients that depend on the empirical data), but rather provides the laws that  govern the dynamics, and that the analysis is based on a time-continuous differential equation that seems to approximate SGD well instead of being applicable to the SGD iterates directly. The validity of the continuous dynamics is demonstrated in numerical results only.  I do agree that a fully predictive framework on SGD directly would be very intersting. Yet, I think the authors are taking important steps towards such a framework, and considering the fact that SGD often behaves surprisingly/unexpectedly (as also stated by R3), I am still quite impressed how accurately the theory matches the actual SGD behavior. For our understanding of how symmetries/invariances in the weights of network architectures influence the training, I believe this paper does provide interesting insights such that I recommend its acceptance. As for a final score, I could go down to a 7 to account for the concerns raised by my fellow reviews, but I think it would mainly reflect my uncertainty about my intuition that a fully predictive analysis on SGD directly might be infeasible, and this aspect should be reflected by the confidence rather than the rating. Thus, I'll give the authors the benefit of the doubt and keep my score, since I really enjoyed reading this paper.

---

> ### Author Response · Authors · 2020-11-18
> **Response to Reviewer 2**
>
> We appreciate that you find our paper very well-written, clear, and elegant without finding any major drawbacks.
>
> **Clarifying our contribution in the light of existing literature.** To help you better understand the novelty and contribution of our work in the light of existing literature, we have added (i) a new discussion in the related work section in the main text and (ii) a new table in section A in the Supplementary Materials presenting how we unify and generalize existing literature on the geometry of loss landscapes through the lens of symmetry. We would also like to thank you for the reference to the work by Li et al. Indeed this work is very relevant and we discuss it in section A of our Supplementary Material how the geometric properties they use in their work fits into our framework of symmetry.
>
> **Addressing your two small questions.** You are correct that (i) loss functions with scale symmetry are discontinuous at the origin, and (ii) that minimizing such a function with L2 regularization under gradient flow does not make sense as you explained.  Yet, we know that any neural network using batch normalization has scale symmetry in the weights and it is common practice to train these networks with weight decay. Furthermore, we do not see this problem in empirics (Figure 5). This discrepancy was the exact issue that motivated us to construct more realistic continuous models of stochastic gradient descent as discussed in section 5.  As we further explained, it is discretization that counteracts the centripetal force of weight decay preventing the weights from collapsing to the origin.  See our updated discussion on this explanation in section 6.

---

### Author Response · Authors · 2020-11-18
**Overview of Paper Update**

We sincerely thank all the reviewers for their positive and constructive feedback that has significantly reshaped our paper. We have incorporated all the suggestions in our updated manuscript, and we hope that the reviewers will reconsider their ratings to reflect the revisions. Below is an overview of the revisions.  Please also see our individual responses to each of the reviewers for the broader context.

## New Figures
Main text:
- Figure 1. Neuron level dynamics are simpler than parameter dynamics.
- Figure 4. Modeling discretization.
- Figure 5. Exact dynamics of VGG-16 on Tiny ImageNet.
- Figure 6. Momentum leads to harmonic oscillation.

Supplementary Materials:
- Figure 11. The planar dynamics of Momentum on VGG-16 on Tiny ImageNet.
- Figure 12. The spherical dynamics of Momentum on VGG-16 on Tiny ImageNet.
- Figure 13. The hyperbolic dynamics of Momentum on VGG-16 on Tiny ImageNet.
- Figure 16. The per-neuron spherical dynamics of Momentum on VGG-16 on Tiny ImageNet.
- Figure 17. The per-neuron hyperbolic dynamics of Momentum on VGG-16 on Tiny ImageNet.

## Major revisions
- Motivating our work in the context of existing literature (section 1, 2)
- Renamed “Inversion Symmetry” to “Rescale Symmetry” (section 3)
- Consistent notation and rigorous proof for gradient/Hessian properties (section 3 and Supplementary Material A)
- An updated section dedicated to commented empirics (section 6)
- New theoretical and empirical study of stochastic gradient descent **with momentum** (see section 5, 6, and Supplementary Material B, E, H)

---

### Decision · Program_Chairs · 2021-01-07
**Final Decision**

**Decision:**

Accept (Poster)

**Comment:**


The paper offers a more systematic treatment of various symmetry-related results in the current literature. Concretely, the invariance properties exhibited by loss functions associated with neural networks give rise to various dynamical invariants of gradient flows. The authors address these dynamical invariants in a unified manner and study them wrt different variants of gradient flows aimed at reflecting different algorithmic aspects of real training processes.

The simplicity and the generality of dynamical invariants are both the strength and the weakness of the approach. On one hand, they provide a simple way of obtaining non-trivial generalities for the dynamics of learning processes. On the other hand, they abstracts away the very structure of neural networks from which they derive, and hence only allow relatively generic statements. Perhaps the approach should be positioned more as a conceptual method for studying invariant loss functions.

Overall, although the technical contributions in the paper are rather incremental, the conceptual contribution of using dynamical invariants to unify and somewhat simplify existing analyses in a clear and clean symmetry-based approach is appreciated by the reviews and warrant a recommendation for borderline acceptance.